# A new *Heterodontosaurus* specimen elucidates the unique ventilatory macroevolution of ornithischian dinosaurs

Viktor J Radermacher[1,2]*, Vincent Fernandez[1,3,4], Emma R Schachner[5], Richard J Butler[1,6], Emese M Bordy[7], Michael Naylor Hudgins[8], William J de Klerk[1,9], Kimberley EJ Chapelle[1,10], Jonah N Choiniere[1]

[1]Evolutionary Studies Institute, University of the Witwatersrand, Johannesburg, South Africa; [2]Department of Earth and Environmental Sciences, University of Minnesota, Minneapolis, United States; [3]European Synchrotron Radiation Facility, Grenoble, France; [4]Natural History Museum, Imaging and Analysis Centre, London, United Kingdom; [5]Department of Cell Biology & Anatomy, School of Medicine, Louisiana State University Health Sciences Center, New Orleans, United States; [6]School of Geography, Earth and Environmental Sciences, University of Birmingham, Birmingham, United Kingdom; [7]Department of Geological Sciences, University of Cape Town, Cape Town, South Africa; [8]Department of Biological Sciences, University of Alberta, Edmonton, Canada; [9]Department of Earth Sciences, Albany Museum, Grahamstown, South Africa; [10]Division of Paleontology, American Museum of Natural History, New York, United States

**Abstract** Ornithischian dinosaurs were ecologically prominent herbivores of the Mesozoic Era that achieved a global distribution by the onset of the Cretaceous. The ornithischian body plan is aberrant relative to other ornithodiran clades, and crucial details of their early evolution remain obscure. We present a new, fully articulated skeleton of the early branching ornithischian *Heterodontosaurus tucki*. Phase-contrast enhanced synchrotron data of this new specimen reveal a suite of novel postcranial features unknown in any other ornithischian, with implications for the early evolution of the group. These features include a large, anteriorly projecting sternum; bizarre, paddle-shaped sternal ribs; and a full gastral basket – the first recovered in Ornithischia. These unusual anatomical traits provide key information on the evolution of the ornithischian body plan and suggest functional shifts in the ventilatory apparatus occurred close to the base of the clade. We complement these anatomical data with a quantitative analysis of ornithischian pelvic architecture, which allows us to make a specific, stepwise hypothesis for their ventilatory evolution.

*For correspondence:
viktorsaurus91@gmail.com

Competing interests: The authors declare that no competing interests exist.

## Introduction

Ornithischia were a morphologically diverse and speciose clade of herbivorous dinosaurs that were major components of terrestrial Mesozoic ecosystems, and whose members include well-known taxa such as *Stegosaurus*, *Triceratops*, and *Parasaurolophus*. Much of the ornithischian body plan is highly derived relative to the morphology of their close dinosaurian relatives, Theropoda and Sauropodomorpha. Although many aspects of the palaeobiology of ornithischians, such as growth strategies (*Redelstorff and Sander, 2009*; *Hübner, 2012*; *Horner et al., 2009*), diets (*Nabavizadeh, 2016*; *Ősi, 2011*; *Nabavizadeh, 2020*), and social behaviour (*Erickson et al., 2009*; *Meng et al., 2004*), have been intensively studied, their respiratory mechanisms remain poorly understood and controversial (*Norman, 2021*).

 

**eLife digest** The fossilised skeletons of long extinct dinosaurs are more than just stones. By comparing these remains to their living relatives such as birds and crocodiles, palaeontologists can reveal how dinosaurs grew, moved, ate and socialised. Previous research indicates that dinosaurs were likely warm-blooded and also more active than modern reptiles. This means they would have required breathing mechanisms capable of supplying enough oxygen to allow these elevated activity levels.

So far, much of our insight into dinosaur breathing biology has been biased towards dinosaur species more closely related to modern birds, such as *Tyrannosaurus rex*, as well as the long-necked sauropods. The group of herbivorous dinosaurs known as ornithischians, which include animals with head ornamentation, spikes and heavy body armour, like that found in *Triceratops* and *Stegosaurus*, have often been overlooked. As a result, there are still significant gaps in ornithischian biology, especially in understanding how they breathed.

Radermacher et al. used high-powered X-rays to study a new specimen of the most primitive ornithischian dinosaur, *Heterodontosaurus tucki*, and discovered that this South African dinosaur has bones researchers did not know existed in this species. These include bones that are part of the breathing system of extant reptiles and birds, including toothpick-shaped bones called gastralia, paired sternal bones and sternal ribs shaped like tennis rackets.

Together, these new pieces of anatomy form a complicated chest skeleton with a large range of motion that would have allowed the body to expand during breathing cycles. But this increased motion of the chest was only possible in more primitive ornithischians. More advanced species lost much of the anatomy that made this motion possible. Radermacher et al. show that while the chest was simpler in advanced species, their pelvis was more specialised and likely played a role in breathing as it does in modern crocodiles.

This new discovery could inform the work of biologists who study the respiratory diversity of both living and extinct species. Differences in breathing strategies might be one of the underlying reasons that some lineages of animals go extinct. It could explain why some species do better than others under stressful conditions, like when the climate is warmer or has less oxygen.

Pulmonary ventilatory systems are highly integrated arrangements, with thoracoabdominal volume change influenced by multiple anatomical regions. Unidirectional airflow is likely a synapomorphy of diapsids (*Schachner et al., 2014*; *Cieri et al., 2014*; *Schachner et al., 2013*; *Farmer and Sanders, 2010*), but the mechanism by which air cycles through the lungs (i.e., ventilation) varies between clades. Interdependent sternocostal movement and visceral displacement drive volume changes in the compliant lungs of extant squamates (*Owerkowicz et al., 1999*; *Brainerd et al., 2015*; *Cieri et al., 2018*) and extant crocodilians (*Claessens, 2009*; *Codd et al., 2019*), whereas sternocostal movement and dorsoventral rocking of the sacrum ventilate the fixed, immobilized lung of birds (*O'Connor, 2004*). Osteological evidence of air sacs or pulmonary diverticula (*O'Connor and Claessens, 2005*; *O'Connor, 2006*; *Wedel, 2006*; *O'Connor, 2009*), and functionally decoupled non-compliant and fixed gas-exchanging regions of the lung (*Wang et al., 2018*; *Perry and Reuter, 1999*; *Schachner et al., 2009*; *Schachner et al., 2011*; *Brocklehurst et al., 2018*) in the major lineages Theropoda and Sauropodomorpha, have led to the modern consensus that most dinosaurs had a 'proto avian-like' respiratory system. Pulmonary anatomy similar to the avian-like respiratory system is also hypothesized to have been present in pterosaurs (*Butler et al., 2009*; *Claessens et al., 2009*), leading to the hypothesis that aspects of proto-avian respiration, including air sacs, are plesiomorphic for the clade Ornithodira (Pterosauria + Dinosauromorpha) (*Wedel, 2006*; *Brocklehurst et al., 2020*).

The aberrant morphology of ornithischian dinosaurs presents a fundamental challenge to this hypothesis. Ornithischians lack aspects of the abdominally mediated breathing apparatus (e.g., gastralia) or the sternocostal apparatus (e.g., mobile sternal ribs) that form key components of the integrated ventilatory systems of other diapsids. Additionally, all known ornithischians lack conspicuous evidence of postcranial skeletal pneumaticity (PSP) that is present in other ornithodiran lineages, indicating that ornithischians did not have pulmonary diverticula that invaded the skeleton

(*Butler et al., 2012*). These observations provide at least two potential hypotheses for ornithischian breathing: (1) these taxa had avian-like air sacs that did not invade the skeleton (similar to some extant diving birds, e.g., *Bucephala clangula*); or, (2) they had a ventilatory strategy entirely divergent from other ornithodirans including living birds. Recently, the lung compliance (the elastic deformation capability of the lung during ventilatory cycles) of ornithischians was reconstructed as uniquely bipartite, with an inflexible, non-compliant anterior portion and a compliant posterior portion (*Schachner et al., 2009*; *Schachner et al., 2011*; *Brocklehurst et al., 2018*). This reconstruction fundamentally differs from reconstructions of other major dinosaurian lineages, which are hypothesized to bear a uniformly non-compliant lung as in living birds that is ventilated via expansion and contraction of anterior and posterior extrapulmonary ventilatory air sacs.

Some previous studies have proposed that ornithischians evolved novel ventilation mechanisms (*Norman, 2021*; *Brett-Surman, 1989*; *Carrier and Farmer, 2000*), but supporting transitional morphological evidence and quantitative comparative analyses are lacking. For example, Brett-Surman (*Brett-Surman, 1989*) argued that the enlarged anterior pubic process (APP) in ornithopods was evidence of the presence of a muscle analogous to the hepatic piston of crocodilians (*M. diaphragmaticus*), and that the elaboration of the APP in hadrosaurs was a subsequent necessity to force air through their intricate and complex narial pathways. Others have suggested instead that non-invasive diverticula characterized ornithischian lineages (*Butler et al., 2012*). However, since the more than 500 known taxa of ornithischian dinosaurs occupied similar habitats (*Zanno et al., 2009*; *Zanno and Makovicky, 2011*), size ranges (*Benson et al., 2014*), and postures as other dinosaur groups, the absence of PSP strongly suggests that they had a lung structure fundamentally different from other ornithodirans.

Since its discovery in 1962 (*Crompton and Charig, 1962*), *Heterodontosaurus tucki* has long been recognized as a taxon crucial for resolving early ornithischian phylogenetic relationships. Most studies seeking to understand ornithischian origins have examined a single articulated skeleton, SAM-PK-K1332, which was fully prepared from its matrix nearly 50 years ago (*Sereno, 1984*; *Sereno, 1986*; *Cooper, 1985*; *Maryanska and Osmólska, 1985*; *Butler et al., 2008*; *Boyd, 2015*). Here, we present a new exquisitely preserved articulated skeleton, AM 4766 (age, stratigraphic provenance, and sedimentological context in Appendix 1). Cautious manual preparation and synchrotron radiation X-ray micro-computed tomography (SRμCT) using an innovative imaging protocol reveal new and unexpected elements of this taxon's anatomy that are not preserved in any other specimens. Much of this new anatomical information has significant bearing on interpretations of the macroevolution of ornithischian respiratory biology. To further our understanding of ornithischian ventilation, we also quantitatively investigated size and shape changes in the evolution of the ornithischian pelvic girdle, paying special attention to the APP. Using a broad sample of ornithischian taxa, we use these data to investigate hypotheses that have implicated the APP in lung ventilation (*Brett-Surman, 1989*; *Carrier and Farmer, 2000*).

## Institutional abbreviations

AM, Albany Museum, Makanda, Eastern Cape, South Africa; NCSM, North Carolina Museum of Natural Sciences, Raleigh, North Carolina; SAM, Iziko South African Museum, Cape Town, South Africa.

## Results

### New anatomy

The osteology of *H. tucki* has been described elsewhere (*Crompton and Charig, 1962*; *Santa Luca et al., 1976*; *Sereno, 2012*; *Galton, 2014*); we focus instead on novel anatomical features preserved in AM 4766 (*Figure 1A*). Visualization of these features was made possible by a high-resolution, phase-contrast enhanced SRμCT with a bespoke reconstruction algorithm developed for this particular specimen (elaborated further in Appendix 1) that has recently been used elsewhere (*Cau et al., 2017*).

### Gastralia

Approximately 18 gastralia are present in total and would have produced two longitudinal rows with each containing 9 gastralia. The gastralia follow the ventral abdominal midline, from the posterior

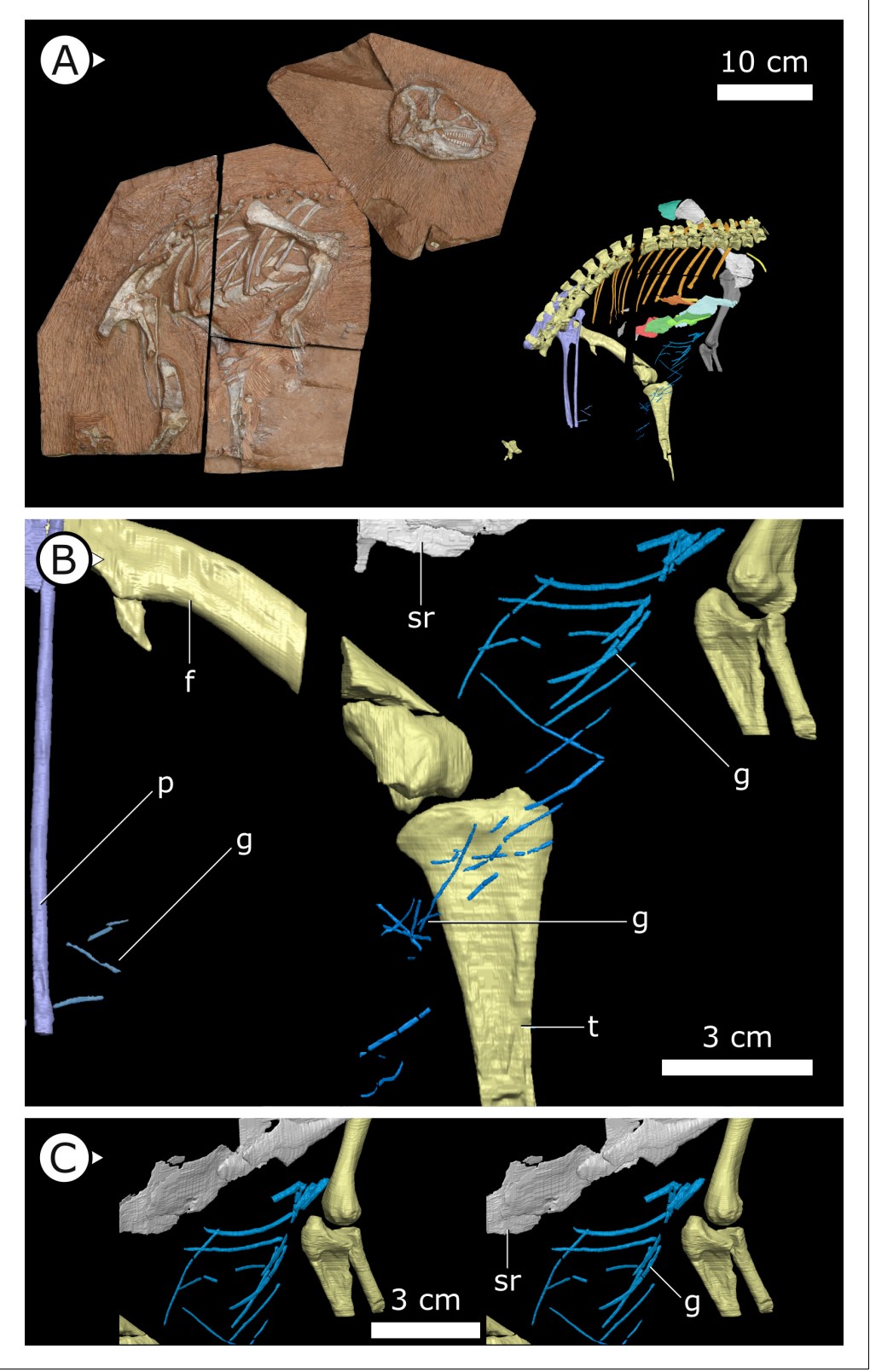

**Figure 1.** Overview of study specimen with emphasis on preserved gastralia. (**A**) Specimen AM 4766 *Heterodontosaurus tucki* on left, with virtual anatomy reconstructed on the right. (**B**) Close-up of gastralia. (**C**) Stereopairs of anterior half of gastralial series. g: gastralia; f: femur (left); t: tibia (left); p: pubis; sr: sternal ribs. Arrows on figure labels point anteriorly.

margin of the sternal plates to the level of the distal ends of the pubes (*Figure 1B*). The first two pairs of gastralia have slightly thickened medial facets that are absent from all subsequent pairs (*Figure 1C*), with the overall thickness of gastralia diminishing posteriorly. The gastralia are autapomorphic among non-avian dinosaurs in lacking a lateral segment (*Claessens, 2004*; *Fechner and Gößling, 2014*; *Barrett et al., 2019*), which is retained in even the diminished gastral basket of early branching avialans (*O'Connor et al., 2015*).

Fragments associated with the *H. tucki* specimen SAM-PK-K1332 are of comparable dimensions to the gastralia in AM 4766; however, they have been removed from context and could potentially represent displaced ossified tendons or posteriormost dorsal ribs. We tentatively identify the long, narrow bone fragments on either side of the proximal femur in the holotype specimen of *Tianyulong confuciusi* STMN 26-3 (*Zheng et al., 2009*; *Appendix 1—figure 3*) as gastralia based on their similarity with AM 4766.

## Sternal plates

Two separate sternal plates are present, although only the left one is complete. The sternal plates are sub-rectangular, their long axes are oriented anteroposteriorly, and they are dorsoventrally thickest on their lateral margin and progressively thin medially (*Figure 2*). The fenestra that perforates the centre of the sternal plate preserved in SAM-PK-K1332, identified by *Sereno, 2012*, is also present in AM 4766. The left sternal plate of AM 4766 bears an autapomorphic, anteromedially projecting, tongue-shaped process that projects abruptly from the anterolateral portion of the sternal plate (*Figure 2B–E*). The proximal portion of this process is partially visible in SAM-PK-K1332, but most of it is still obscured by matrix. The exact nature and function of the tongue-shaped process is currently unknown, but, when paired, they likely buttressed the region between coracoids. The posterolateral corner of the sternal plate of SAM-PK-K1332 has a small but distinct protuberance, identified as an articulation for the sternal ribs (*Sereno, 2012*), herein referred to as a costal process (*Figure 2C, D*). While this structure appears to be missing from AM 4766, its absence cannot be confidently confirmed as the resolution in this region of the SRµCT data is diminished by metallic inclusions obscuring boundaries between bones and matrix.

## Sternal ribs

Three pairs of sternal ribs are preserved in AM 4766, each similar in size and morphology (*Figures 3* and *4*). The sternal ribs have a spatulate morphology, with an elongate and semi-cylindrical anterior half, and an abruptly dorsoventrally expanded posterior half that thins to a mediolaterally compressed, sheet-like structure, similar to the avialan *Jeholornis prima* (*Zheng et al., 2020*; *Figure 4G*). A thickened nub on the posterior apex of this sheet-like portion forms a monocondylar sternocostal articulation with the distal end of the corresponding dorsal rib. A similar articular relationship is present between the sternal, intermediate, and dorsal ribs of extant crocodilians (*Claessens, 2009*; *Brocklehurst et al., 2017*). Although the gross morphology of their sternal ribs differs, the sternal and dorsal ribs of pterosaurs (e.g., *Rhamphorhynchus muensteri*, *Figure 4F*) also bear monocondylar sternocostal joints that are strikingly similar to those of *H. tucki* (*Claessens et al., 2009*). The sternal ribs of AM 4766 contrast markedly with the few other ornithischian examples: for example, in *Thescelosaurus neglectus* (NCSM 15728) (*Figure 4E*) and *Nanosaurus agilis* (BYU 163) (identified as 'costal cartilage' in *Carpenter and Galton, 2018*), the sternal ribs are comparatively shorter, subrectangular, and bearing broad butt joints rather than condylar articulations at their distal and proximal ends.

## Clavicles

The paired clavicles (*Figure 5A, B*) are preserved in life position anterior to the scapulocoracoid and are proportionally long, thin, and bowed posteriorly. The proximal end of the left clavicle is marginally thicker than the rest of this element and gently tapers laterally. Among ornithischians, clavicles are mostly known in basal ceratopsians and neoceratopsians from the Cretaceous, for example, *Psittacosaurus mongoliensis* (*Fairfield, 1924*; *Sereno, 1990*), *Psittacosaurus sibiricus* (*Averianov et al., 2006*), *Auroraceratops rugosus* (*Morschhauser et al., 2018a*), *Leptoceratops gracilis* (*Sternberg, 1951*), *Montanoceratops cerorhynchos* (*Chinnery and Weishampel, 1998*), and *Protoceratops andrewsi* (*Brown and Schlaikjer, 1940*) but are also present in the basal thyreophoran *Scelidosaurus*

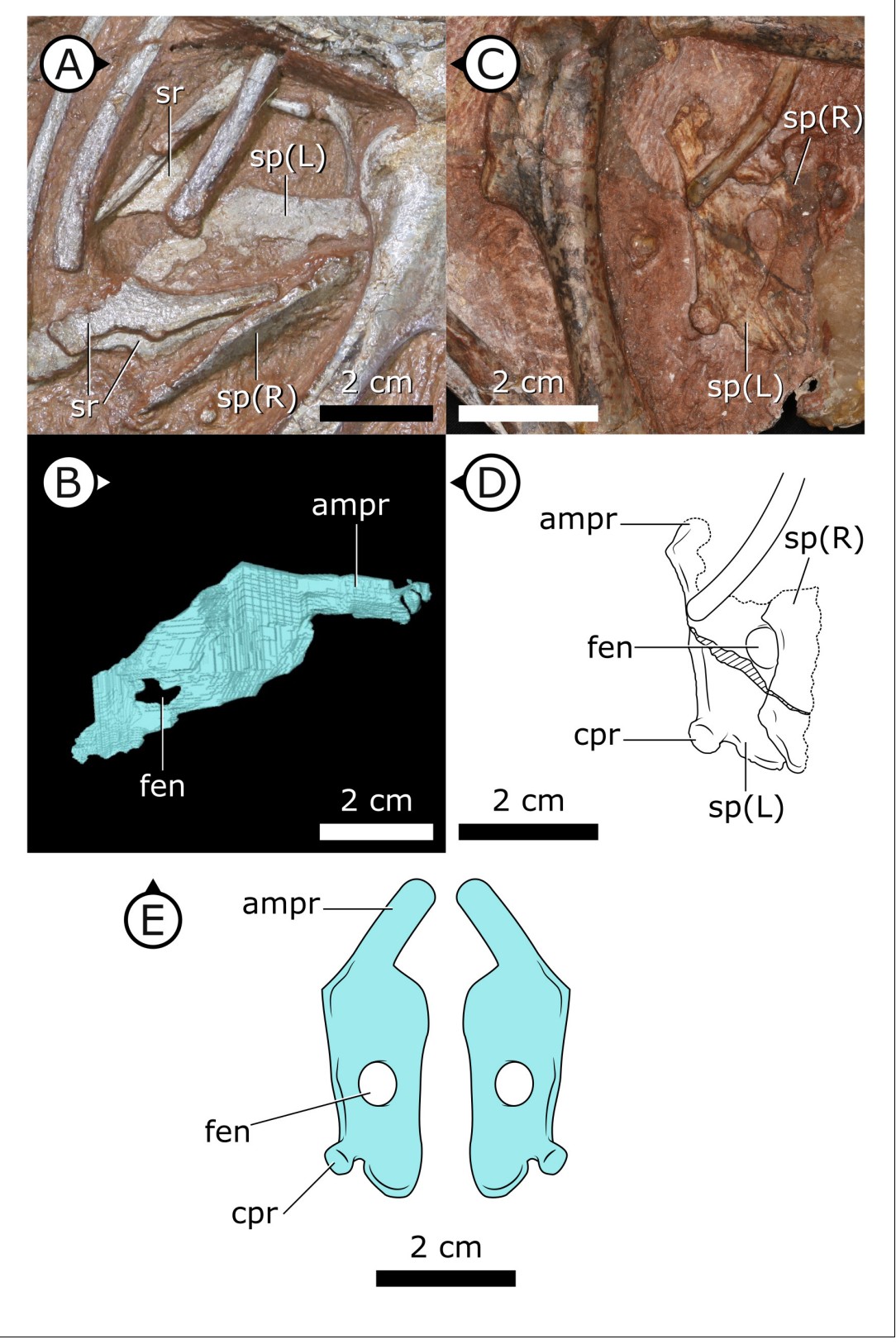

**Figure 2.** Sternal plates of *H. tucki*. (**A**) Location of sternal plates in AM 4766, (**B**) segmented left sternal plate of AM 4766, and (**C**) sternal plates in SAM-PK-K1332. (**D**) Line drawing of sternal plates in SAM-PK-K1336. (**E**)
*Figure 2 continued on next page*

*Figure 2 continued*

Composite line drawing of *H. tucki* sternal plate anatomy informed by both specimens. ampr: anteromedial process; cpr: costal process; fen: fenestra; sp: sternal plate. Arrows on figure labels point anteriorly.

*harrisonii* (**Norman, 2020**) as well as a new, undescribed taxon that is purported to be at the base of Ornithopoda (**Spencer et al., 2020**). The clavicles of *H. tucki* are similar to those of ceratopsians in contouring the anterior margin of the scapulocoracoid, with no apparent contact present between the clavicles along their length. The clavicles of AM 4766 differ from those of ceratopsians in overall size. AM 4766 has clavicles that are ~60% the length of the anterior margin of the body of the

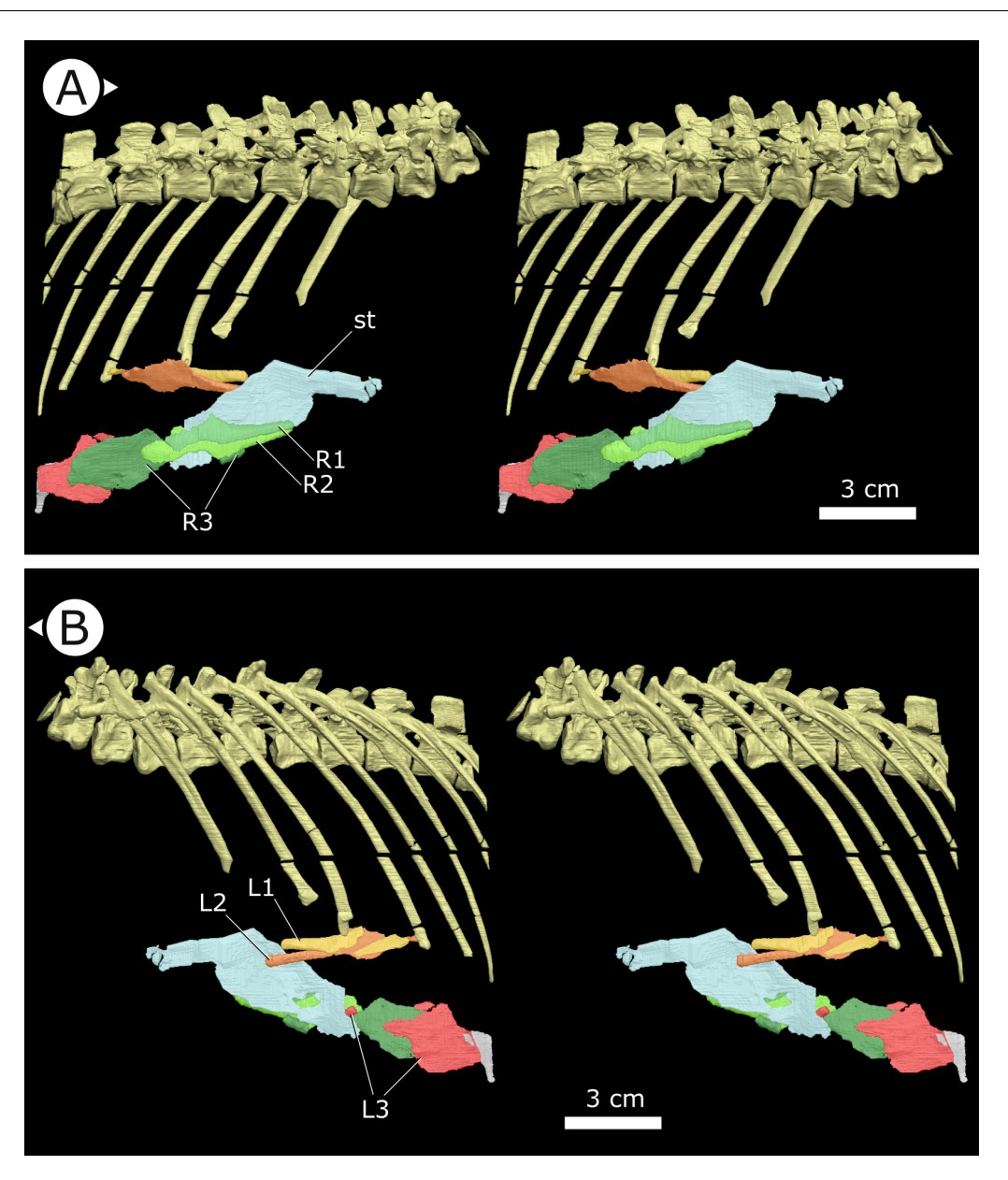

**Figure 3.** Stereopairs of segmented sternal ribs preserved in AM 4766. (**A**) Right lateral view and (**B**) left lateral view. st: sternal plate (left); L/R 1/2/3: left/right first, second, and third sternal ribs (anterior to posterior). Arrows on figure labels point anteriorly.

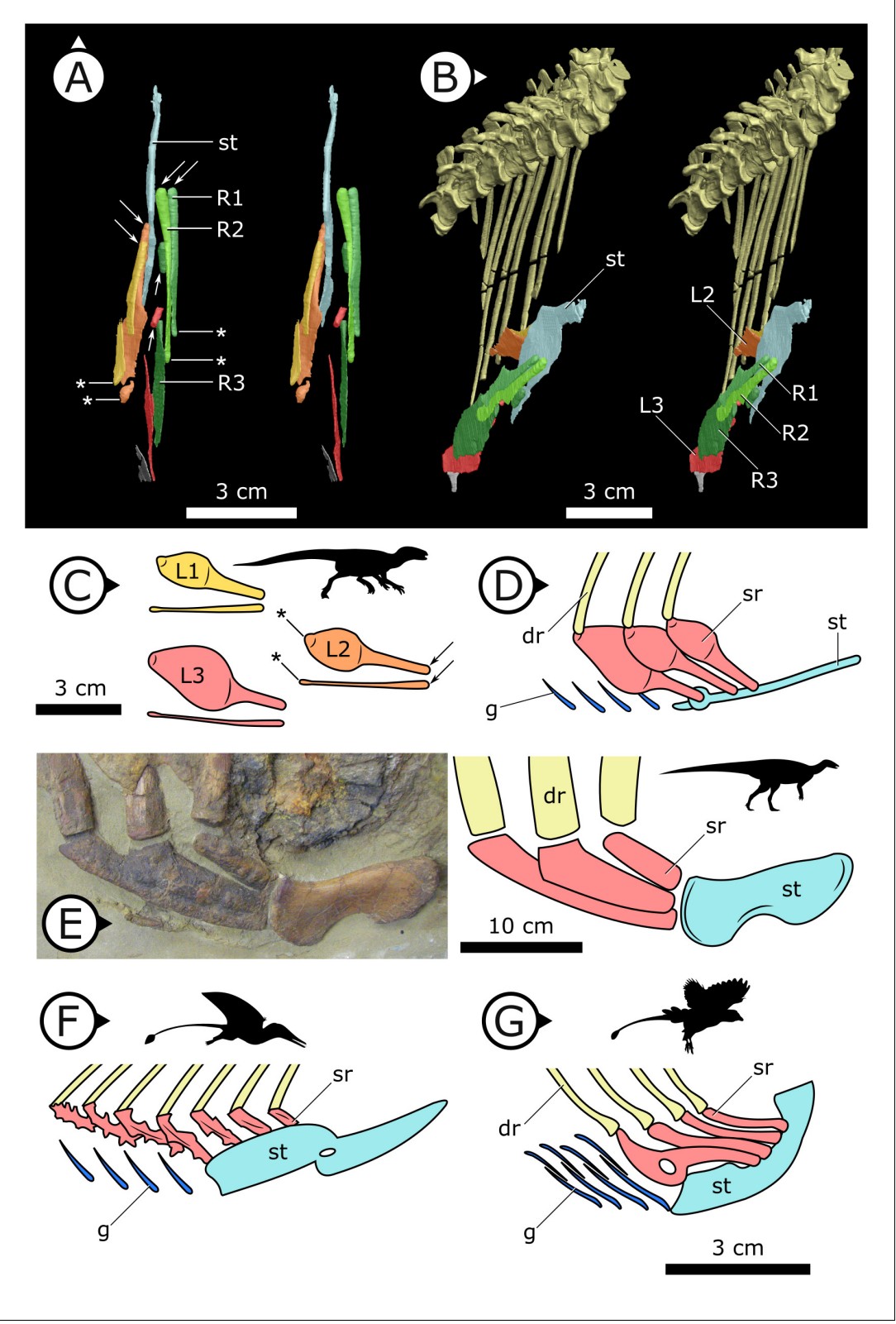

**Figure 4.** Comparative details of sternal ribs in other ornithodirans. (**A**) Stereopairs of AM 4766 in (left) dorsal and (right) anteroventrolateral views. (**B, C**) Idealized version of sternal ribs present in AM 4766. (**D, E**) Photo and line drawing of sternal complex in *Thescelosaurus neglectus* (NCSM 15728). (**F**) Schematic sternal complex of *Rhamphorhynchus*, modified from ***Claessens et al., 2009***. (**G**) Schematic sternal complex of *Jeholornis*, modified

*Figure 4 continued on next page*

*Figure 4 continued*

from *Zheng et al., 2020*. Arrows and asterisks point to sternal and dorsal rib articulation points, respectively. dr: dorsal ribs; g: gastralia; L/R 1/2/3: left/right first, second, and third sternal ribs (anterior to posterior) of AM 4766; sr: sternal ribs; st: sternal plates. Arrows on figure labels point anteriorly.

scapulocoracoid (i.e., excluding the scapular blade), where the clavicles of ceratopsians are approximately 40% of the length of the anterior margin of the scapulocoracoid.

## Suprascapula

The suprascapula (*Figure 5*) is sub-trapezoidal in shape, being broader distally than it is proximally. The distal margin of the suprascapula is concave and articulates with the proximal, convex margin of the scapular blade. A suprascapula is also present in SAM-PK-K1332 and was originally described by *Santa Luca et al., 1976* as a 'cartilaginous extension' that capped the dorsal margin of the scapula.

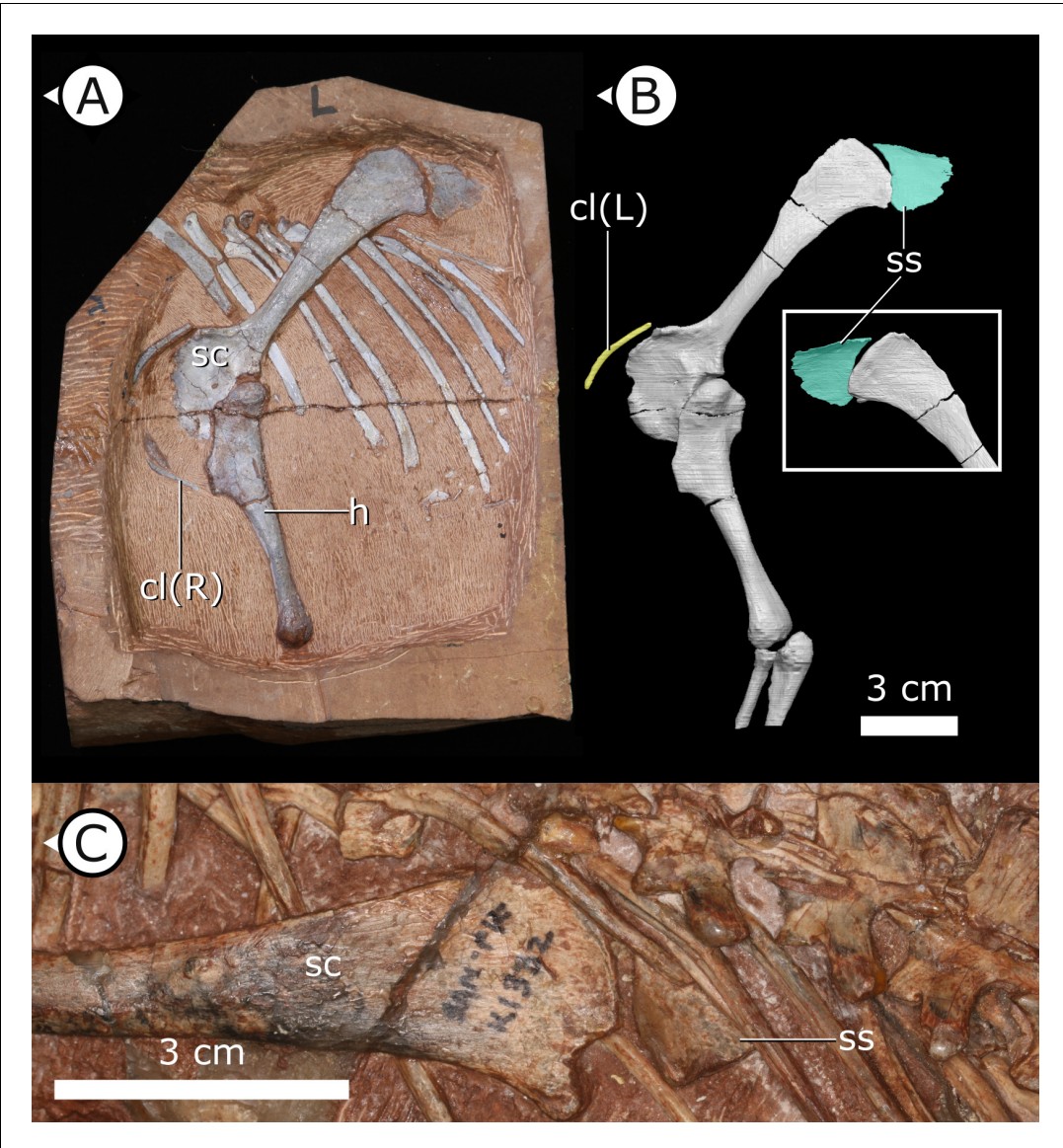

**Figure 5.** Accessory ossifications of pectoral girdle in *H. tucki*. (A, B) Clavicles and suprascapula of AM 4766 (B is segmentation of µCT data); (C) suprascapula in SAM-PK-K1332. cl (L/R): left/right clavicle; h: humerus; sc: scapula; ss: suprascapula. Arrows on figure labels point anteriorly.

At present, we are unable to eliminate the possibility that this structure is indeed cartilaginous, but it is undistorted and has clearly defined margins that allow us to tentatively consider the suprascapula as an ossification (rather than a chondrification). An ossified suprascapula has only been described in one other dinosaurian taxon, the Cretaceous neornithischian *Parksosaurus warreni* (*Sternberg, 1936*).

## Internal thoracic ceiling and vertebral structure

Synchrotron scanning of AM 4766 permitted the reconstruction of vertebral morphology previously obscured in specimen SAM-PK-K1332. Notably, the parapophyses migrate anterodorsally as the vertebral series progresses posteriorly (*Figure 6*): on the first, second, and third dorsal vertebrae, the parapophyses are located immediately ventral to the diapophyses; the fourth and fifth dorsal vertebrae mark the transition where the parapophyses migrate dorsally from the centrum and onto the neural arch; and from the sixth dorsal vertebrae and in all more posterior dorsals, the parapophyses are anterior to the diapophyses and on the same horizontal level. Our reconstruction based on SRµCT data shows that the entire internal structure of the cervical, dorsal, sacral, and proximal caudal vertebral column lacks pneumatic chambers or fossae, including those areas often implicated in the early evolution and development of PSP (*Wedel, 2006*; *Butler et al., 2009*; *Benson et al., 2012*), conclusively showing that early ornithischians lacked PSP.

## Quantitative analysis of ventral pelvic architecture

Our measured variables show strong ($\log_{10}$ pubic rod length; $r^2 = 0.858$) to very strong ($\log_{10}$ APP length, ischial length; $r^2 > 0.96$) correlations with body size (represented in our analysis by $\log_{10}$ femur length, see Materials and methods, and *Appendix 1—figures 4* and *5*). We therefore corrected for phylogenetic and allometric effects by using the residuals of phylogenetically corrected generalized least squares (pGLS) regressions of each variable against $\log_{10}$ femoral length. The residuals from our pGLS regression of each of our three variables showed poor correlation with $\log_{10}$ femoral length. This indicates that changes in pubic and ischial dimensions are largely dissociated from the allometric effects of body size (see *Appendix 1—figure 5A–F*).

Optimizing residuals on the phylogeny (*Figure 7*) shows that later branching taxa have APPs that are elongate relative to early branching taxa. APP elongation occurs at the base of Genasauria, and within this clade it is modified comparatively little over its subsequent history. There are generally declining rates of change in APP length in later-branching lineages and temporally later-appearing tips of the tree, with exactly zero known instances of reversion to the plesiomorphic relative length. Derived hadrosaurs and neoceratopsians apparently appear to have slightly shorter APPs relative to earlier-diverging taxa of their respective clades; however, it should be noted that these taxa dorsoventrally expand the APP independently, significantly increasing the surface area for muscle attachment.

Early branching ornithischians have long pubic rods, which subsequently shorten independently in ornithopods and marginocephalians well after the APP begins to elongate on the tree (i.e., after the major splits in Genasauria). Ischial length shows a more complex pattern, with most ornithischians retaining the plesiomorphic proportional length, but with stegosaurs showing large decreases in relative length and ornithopods and certain ceratopsians showing modest increases.

We used each set of residuals as continuous characters for an evolutionary model testing analysis using phylogenetic comparative methods (see Materials and methods). Among these, 'Early Burst' is strongly preferred for the evolution of APP length and performs better than other competing models (Akaike Information Criterion [AICc] weight: 99.99%, likelihood ratio test p<0.01; see *Table 1*). The 'Early Burst' model posits declining evolutionary rates over time, that is, expected variance is higher between earlier-branching taxa (*Harmon et al., 2010*), matching our qualitative observations from mapping residuals on the tree (*Figure 7*). Pubic rod length is best modelled by a 'Drift' model (AICc weight: 82.34%, p=0.01), and 'Stasis' is more strongly, but non-significantly preferred for ischial length (AICc weight: 59.37%; p=1).

## Discussion

### Gastralia and their implications

The nearly sequentially complete gastral basket of AM 4766 is the first known in ornithischian dinosaurs, and the tentative identification of gastralia in the holotype of the Chinese taxon *T. confuciusi* suggests that gastralia may have been present in all heterodontosaurids. With gastralia being plesiomorphically ubiquitous across a range of tetrapod clades, discovering gastralia in Heterodontosauridae is not surprising as this clade is consistently recovered as the earliest branching lineage of ornithischian dinosaurs (*Butler et al., 2008*; *Boyd, 2015*). It is more surprising, however, that these gastralia are retained in *H. tucki* despite its typical ornithischian retroverted pubis. Three-dimensional reconstruction of our SRμCT data clearly demonstrate gastralia in close association with the distal-most point of the pubes, indicating that their complete retroversion (opisthopuby) was achieved with the gastralia still intimately coupled. Together, these observations contest previous hypotheses that reasoned that a divorce of the gastral basket from the pubis was a necessary prerequisite for ornithischian pubic retroversion (*Rasskin-Gutman and Buscalioni, 2001*). Furthermore, the association of the gastralia with the distal end of the pubic rod indicates that the latter structure is homologous to the pubic shaft/apron of other archosaurs (*Galton, 1970*, contra references therein), and that the APP is a de novo feature.

### Sternal ribs and their function

The presence of sternal ribs in *H. tucki* extends the occurrence of these bones from late diverging taxa like *T. neglectus* and other relatively late-branching, small-bodied Late Jurassic and Cretaceous neornithischians (*Carpenter and Galton, 2018*; *Butler and Galton, 2008*) to the basalmost members of Ornithischia. This broader distribution strongly implies that the presence of sternal ribs may optimize as an ornithischian plesiomorphy. However, the sternal ribs we describe in *H. tucki* are autapomorphic in morphology, differing markedly from those of other ornithischians, and showing clear evidence of being mobile about their dorsal rib and sternal plate joints (*Figure 4A–C*). The dorsoventrally expanded dorsal and ventral margins of these ribs were likely attachment sites for intercostal musculature and in this way perhaps analogous to similar projections (sternocostapophyses) on the sternal ribs of pterosaurs (*Claessens et al., 2009*), the uncinate processes of maniraptorans (*Tickle et al., 2012*; *Codd et al., 2008*), and the remarkably similar sternal ribs of the ornithothoracine *J. prima* (*Zheng et al., 2020*) – all of which are adaptations hypothesized to increase lever-arm potential and facilitate efficient deformation of the body wall to drive ventilation.

### Sternum

The complex sternal plates of AM 4766 are distinct from the comparatively simple 'hatchet-shaped' sternal plates of iguanodontians and the 'kidney-shaped' (reniform) sterna of other neornithischians but are not unique among Dinosauria. Instead, the complex sternal plates of AM 4766 bear similarities with early- and late-diverging theropods such as *Tawa hallae* (*Bradley et al., 2019*) and various avialans (*Zheng et al., 2012*; *O'Connor et al., 2015*), respectively. Features like the tongue-shaped process of AM 4766 and the coracoid facet of *T. hallae* are strikingly similar in their dimensions, location, and abrupt change in orientation relative to the posterior half of their respective sternal plates. Further similarities include the single knob-like costal process of AM 4766 exhibiting a similar morphology to the series of costal processes of *T. hallae*, and the analogous position of the lateral tubercula of enantiornithines (*Zheng et al., 2012*). It is unclear whether these sternal similarities are homologous, but they are likely functionally analogous.

### Quantitative analysis of pelvic evolution

The nature of change in the relative length of the APP is conspicuous from both qualitative and quantitative analyses of pelvic evolution. Innovation in APP length occurred early in ornithischian evolution, before the diversification of genasaurians, and after this significant early burst (see *Figure 7*, *Table 1*), modifications of the APP were generally restricted to gross shape differences that do not affect the relative length: derived ornithopods evolved a large, lobate APP with derived neoceratopsians evolving an APP that fanned-out anteriorly. We interpret these results as rapid

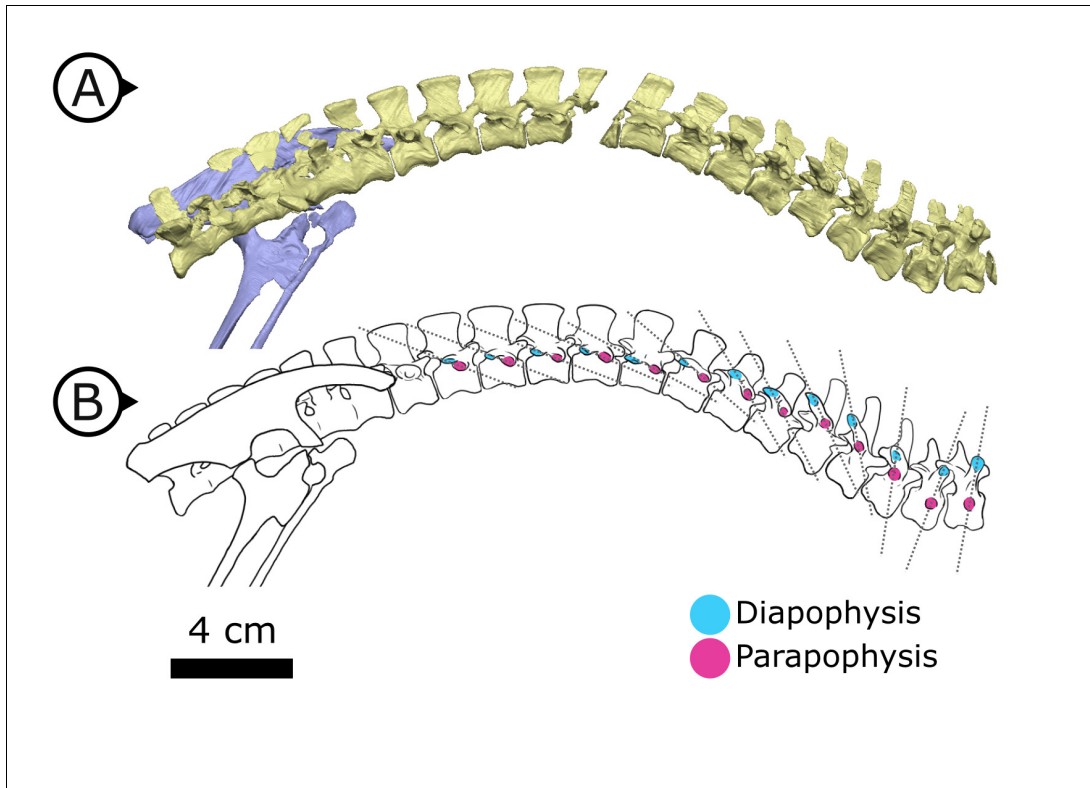

**Figure 6.** Changing diapophyseal and parapophyseal relationships in AM 4766. (**A**) Virtual reconstruction of cervicothoracic, thoracic, and sacral vertebrae of AM 4766; (**B**) Line drawing of (**A**) with diapophyses and parapophyses colour-coded in cyan and magenta respectively. Dashed lines indicate shifting position of parapophyses relative to the accompanying diapophyses. Arrows on figure labels point anteriorly.

switching of optimal phenotypes for APP size; from the plesiomorphically small condition in *H. tucki* to a proportionally long APP that is then maintained across all later-branching ornithischian lineages.

We consider this as evidence that the APP was involved in a major macroevolutionary shift in ornithischian dinosaurs, which occurred immediately prior to their radiation in the Jurassic. Moreover, the timing of this change in the relative length of the APP cooccurs with the reduction and loss of the gastralia and possibly the loss or reduction of sternocostal mobility. These results are consistent with, but greatly expand upon, Brett-Surman's hypothesis (*Brett-Surman, 1989*) that the enlarged APP is an adaptation of ornithischians involved in driving considerable changes in abdominothoracic volume.

The pelves of published ankylosaur specimens are often obscured or incomplete and do not contain sufficient measurement information to include in this analysis (*Kirkland and Carpenter, 1994*; *Carpenter et al., 2013*; *Arbour and Currie, 2013*; *Xu et al., 2001*). Nevertheless, we observe that despite highly derived pubic rod morphologies, including near loss in late branching taxa like *Euoplocephalus tutus*, the APP is retained as a process fused to the ventral surface of the ilium (*Carpenter et al., 2013*). This strongly implies that the APP was subject to a constraint that favoured its retention when the rest of the pubis was made redundant.

This pattern of pubic evolution is best explained by a 'Drift' evolutionary hypothesis and suggests a trend away from elongate pubes. The reduction of the pubic rod is pervasive in ornithischians as it is independently lost in derived iguanodontians, neoceratopsians, and pachycephalosaurs (as well as ankylosaurs; *Carpenter et al., 2013*). This likely signifies a relaxing of a plesiomorphic constraint between the hypaxial abdominal musculature and the pubis. This reduction of the pubic rod is decoupled temporally and phylogenetically from the loss of gastralia and the expansion of the APP,

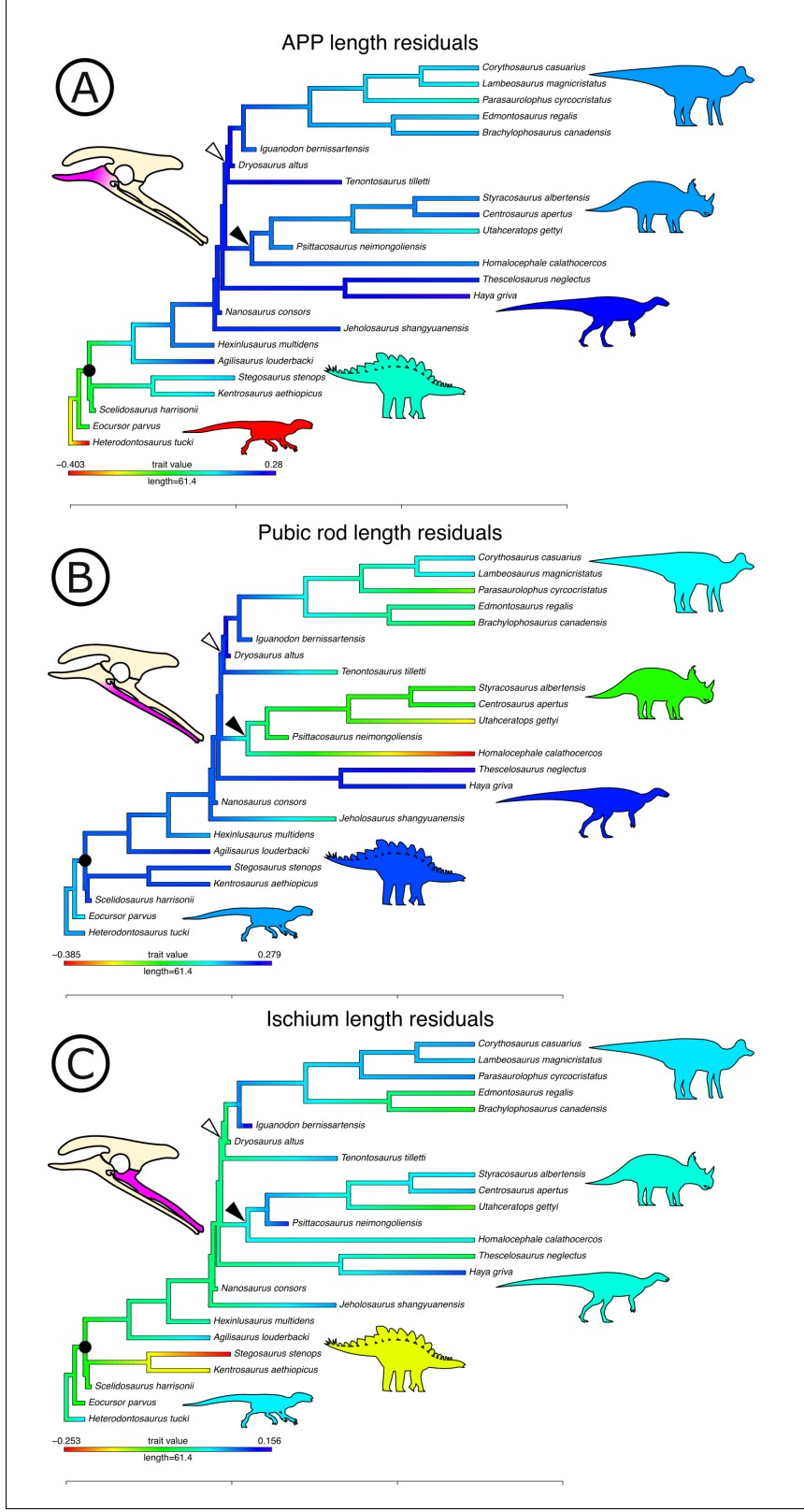

**Figure 7.** Phylogenetically corrected results of ornithischian pelvic element analysis. Phylogenetically corrected generalized least squares residuals results of (**A**) evolution of the anterior pubic process, (**B**) pubic rod, and (**C**) ischial length. Closed circle, Genasauria; open triangle, Ornithopoda; closed triangle, Marginocephalia.

*Figure 7 continued on next page*

*Figure 7 continued*

Silhouettes represent (from left to right): *Heterodontosaurus*, Stegosauria, Parksosauridae, Neoceratopsia, and Hadrosauridae.

strongly suggesting that the pubic rod was rendered vestigial. It is possible that modifications like the bowed ischium of neoceratopsians and the 'ischial boot' of some hadrosaurs are responses to the ischium subsuming the myological role previously played by the pubis.

Ischial residuals are harder to interpret, and despite being best explained by a 'Stasis' model, there is no statistical significance between this explanatory model or any other evolutionary models we tested. Qualitatively, most ornithischian dinosaurs have similar-length ischia relative to their body size. The most conspicuous departures from this are in late-branching ornithopods, where the ischium is elongated (i.e., strong positive residuals), and in stegosaurs where the ischium is shortened (i.e., strong negative residuals). That the pubic rods of stegosaurs remained elongate and the ischia short and robust almost certainly indicates that some other selective pressure such as tail-driven defence (*Mallison, 2011*; *Carpenter et al., 2005*) was imposed on the ischium that prevented it from supplanting the role of the pubis in anchoring abdominal musculature. Nevertheless, both ischial modifications occur temporally and phylogenetically well after the increase in relative APP length and the loss of gastralia.

## A new model of lung ventilation in ornithischian dinosaurs

The anatomical features presented here provide consilient evidence that *H. tucki* preserves morphologies that reflect early steps in the evolution of a novel means of lung ventilation in ornithischian dinosaurs. Below, we review this evidence and propose a potential model for the ornithischian ventilation system.

### Gastralial modification

Gastralia are widespread among Palaeozoic and Mesozoic tetrapods but have never before been unambiguously reported in Ornithischia. Among other dinosaurian lineages, theropods retain their gastral basket until the evolution of neornithine birds, and sauropodomorphs lose their gastralia relatively late in their evolutionary history at the base of Eusauropoda – potentially retaining gastralia even during the emergence of Neosauropoda (*Tschopp and Mateus, 2013*).

The reduction or loss of the gastralia independently occurs in other major tetrapod lineages like stem turtles (*Lyson et al., 2013*; *Schoch and Sues, 2020*), and eutheriodont therapsids including mammals (*Cisneros et al., 2015*). Interestingly, specialized lung ventilatory mechanisms are present in all extant clades that have lost (birds, mammals) or co-opted the gastralia (turtles). Some workers have explicitly linked the loss of gastralia and subsequent thoracic and lumbar differentiation of therocephalian and cynodont therapsid axial skeletons to the evolution of a mammalian-style, diaphragm-driven ventilatory arrangement (*Perry et al., 2010*; *Brink, 1956*).

The specialized facets on the medial gastralial elements of non-avian theropods have led multiple authors (*Carrier and Farmer, 2000*; *Claessens, 2004*; *Fechner and Gößling, 2014*; *Codd et al., 2008*; *Lambe, 1917*) to hypothesize that gastralia played an important role in ventilating the lungs, a mechanism *Carrier and Farmer, 2000* termed 'cuirassal breathing' that was inherited from a common non-dinosaurian archosaurian ancestor. These hypotheses posit the gastralia would have

**Table 1.** Akaike Information Criterion weights and likelihood ratio test (p) statistics for the evolutionary models analysed here (see Materials and methods).
Bold values indicate preferred explanatory model for each measured pelvic variable. Likelihood ratio tests are between the preferred model and the next most preferred model. BroMo: Brownian Motion; OU: Ornstein–Uhlenbeck.

|  | BroMo (%) | OU (%) | Early-burst (%) | Drift (%) | Stasis (%) | p = |
|---|---|---|---|---|---|---|
| APP length | 0.00 | 0.00 | **99.99** | 0.00 | 0.00 | $1.33^{-07}$ |
| Pubis length | 11.06 | 3.56 | 2.97 | **82.34** | 0.00 | 0.01 |
| Ischium length | 5.21 | 25.24 | 8.73 | 1.45 | **59.37** | 1.00 |

facilitated expansion and contraction of the body wall to facilitate volumetric changes in the thoracoabdominal cavity (*Carrier and Farmer, 2000*; *Claessens, 2004*; *Codd et al., 2008*; *Lambe, 1917*). Although in extant crocodilians the gastralia themselves only contribute a relatively small amount to such volumetric changes in isolation (*Claessens, 2009*), the gastral basket is integral in bridging the sternocostal complex and mobile pubis, serving as an attachment site for muscles fundamental to body wall deformation and function of the 'hepatic piston'.

## The archosaurian pelvis as a respiratory locus

In archosaurian ventilation models, the involvement of the pelvis is ubiquitous, ranging from pelvic rocking in birds (*Baumel et al., 1990*), to the hepatic piston in crocodilians (*Farmer and Carrier, 2000*), to the prepubis of pterosaurs (*Claessens et al., 2009*). Anterior bony projections of the pubic region are key components of these models, including the mobile pubis of crocodilians, and the prepubis and puboiliac complex in pterosaurs. *Carrier and Farmer, 2000* highlighted the APP as the integral locus for interpreting ornithischian lung ventilation, focusing their hypothesis on the major genasaurian clades Neoceratopsia, Ornithopoda, and Stegosauria. *Macaluso and Tschopp, 2018* hypothesized that pubic retroversion in dinosaurs is linked to the evolution of an innovative ventilatory mechanism, arguing that the plesiomorphic cuirassal breathing proposed by *Carrier and Farmer, 2000* constrains the pubis into the propubic condition, and that evolution of mesopubic and opisthopubic conditions indicates a relaxing of those constraints as a new mechanism evolves.

The role of the APP in ventilation has been contentious, however, with other authors assigning it a locomotory function (as the origin of the *ambiens* [*Maidment and Barrett, 2011*] or *pubotibialis* [*Galton, 1969*] muscles). Although the locomotory and ventilatory explanations are not mutually exclusive, the evidence gathered here makes us consider the locomotory role to be a poor explanation for APP changes for the following reasons. First, our evolutionary analysis clearly shows that the major changes in the APP residuals are phylogenetically and temporally divorced from the independent acquisitions of quadrupedality and major postural changes (*Maidment and Barrett, 2011*; *Maidment and Barrett, 2012a*; *Maidment and Barrett, 2012b*; *Barrett and Maidment, 2017*). Second, although there is little available data from extant taxa, the *ambiens* muscle appears to have weak negative allometry in *Dromaius novaehollandiae* (*Lamas et al., 2014*), suggesting that body size increases in ornithischian lineages would not drive a trend of disproportional APP increase (additionally, our analysis of pubic measurements using residuals precludes this). Third, the APP is subparallel to the vertebral column and medial to the ribcage, thus precluding it from being a major driver of hindlimb extension or retraction when body wall musculature is reconstructed (principally *M. obliquus abdominus externus*; *Fechner and Gößling, 2014*; *Fechner and Schwarz-Wings, 2013*). These lines of evidence together indicate that the factors driving the evolution of APP length and shape are distinct from locomotory influences.

## The pelvic bellows

In total, our observations here show that *H. tucki* has reduced gastralia, an apomorphically elaborate sternum, well-developed and mobile sternal ribs, an incipient APP, and completely lacks PSP. We propose a single explanatory model for these observations: that *H. tucki* is a transitional animal preserving the early steps in the evolution of a unique ventilation mechanism in ornithischian dinosaurs. We name this model the 'pelvic bellows' and elaborate on it below. This model does not require us to make *ad hoc* assumptions about airflow direction (i.e., unidirectional versus tidal), but phylogenetic bracketing predicts intrapulmonary unidirectional airflow in Ornithischia and is fully compatible with our model (*Schachner et al., 2014*; *Cieri et al., 2014*; *Farmer and Sanders, 2010*; *O'Connor and Claessens, 2005*).

Stem dinosauromorphs (*Figure 8A*; *Kammerer et al., 2020*) (or stem sulcimentisaurians; *Müller and Garcia, 2020*) bear the plesiomorphic archosaurian condition of a typical gastral basket connecting to a propubic pelvis, and they lack both PSP (*Butler et al., 2009*) and an APP. This points to cuirassal ventilation as the primary means of volume change. Interestingly there is evidence of a bipartite, semi-compliant lung in *Silesaurus opolensis* (*Schachner et al., 2011*), adding potential support to a hypothesized but controversial relationship between silesaurids and Ornithischia (*Müller and Garcia, 2020*; *Ferigolo and Langer, 2007*).

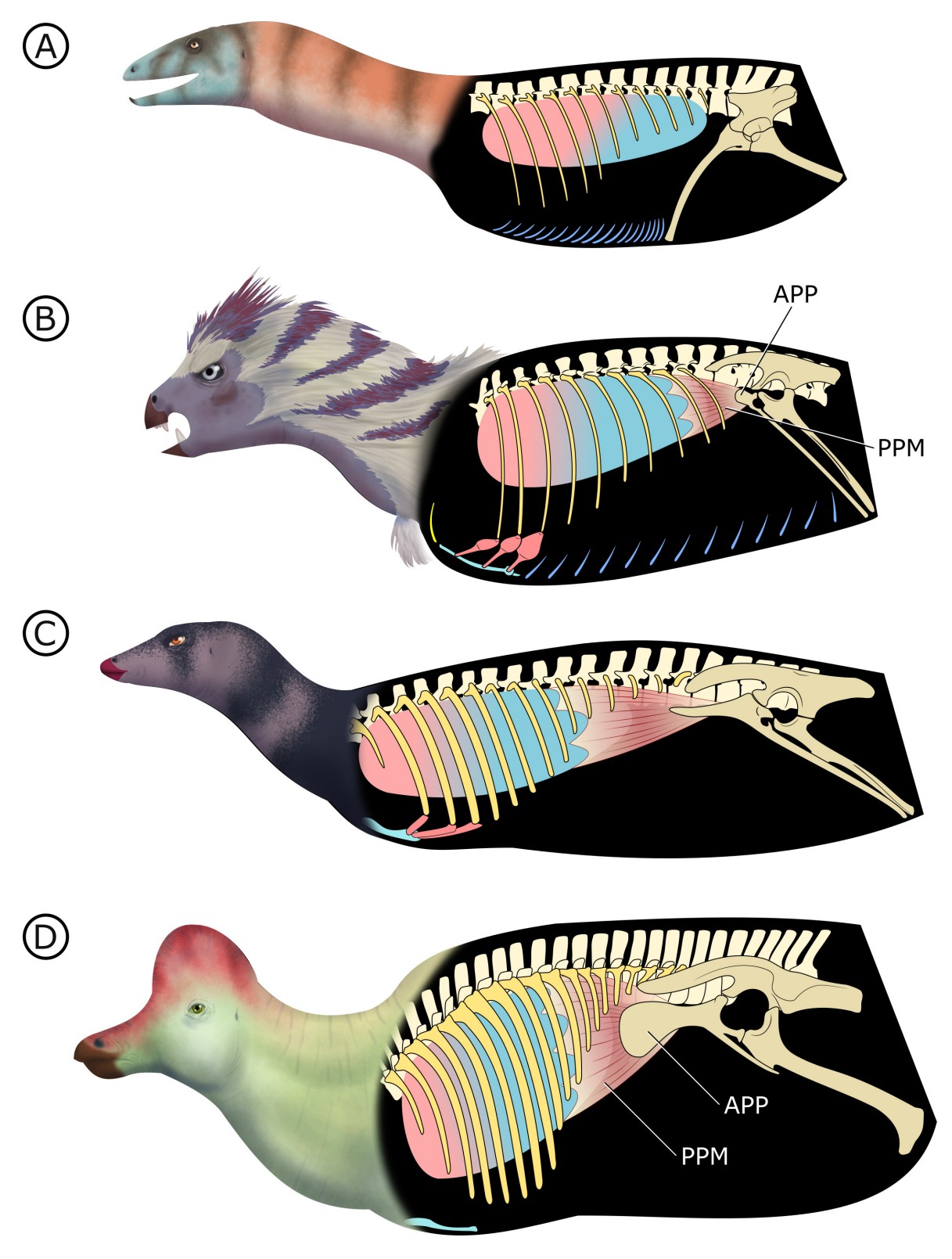

**Figure 8.** Hypothesized stepwise evolution of the ornithischian pelvic bellows and accompanying skeletal modifications and myological innovations. (A) Silesaurus (outgroup), (B) Heterodontosaurus, (C) Thescelosaurus, to (D) Corythosaurus. Lung size is an approximation; red and blue portions of the lung represent hypothetical reconstructions of non-compliant and compliant lung regions, respectively. APP: anterior pubic process; PPM: puboperitoneal muscle. Not to scale.

In early branching ornithischians, exemplified by *H. tucki* (*Figure 8B*), the retroverted pubes and the reduced gastralia indicate that the cuirassal breathing mechanism (*Carrier and Farmer, 2000*) is still present but has reduced capacity to affect changes in volume. The sternal ribs of *H. tucki* would have facilitated the pivoting and leveraging of the sternum and aided in its posteroventral contraction, providing substantial volumetric changes. The small APP would have served as a nascent area for the origination of a muscle analogous to the dorsal component of *M. diaphragmaticus* in living crocodilians, which we term the 'puboperitoneal muscle'. We hypothesize that the puboperitoneal muscle would have functioned as an accessory lung ventilator in early ornithischians, similar to the accessory ventilatory function provided by the *iliocostalis* musculature of some crocodilians (*Codd et al., 2019*). The puboperitoneal muscle would have provided an additional anteroposterior vector to the dorsoventral displacement already afforded by the cuirassal and sternocostal mechanisms. Upon inspiration, contraction of *M. rectus abdominus* would have distended the gastral basket and the sternal complex posteroventrally, the puboperitoneal musculature simultaneously contracting to generate negative pressure in the posteriorly compliant half of the lung. During expiration, the abdomen, sternal complex, and puboperitoneal muscle relaxed and would rebound anterodorsally to force air out.

Gastralia are seemingly lost amongst early branching ornithischians, and the cuirassal breathing mechanism is no longer present in Genasauria (*Figure 8C*). However, the pubic rod is still long and *M. rectus abdominus* is likely still present, not entirely precluding the possibility of body wall deformation by contractions of hypaxial musculature. The sternal ribs are greatly simplified or lost. When they are present, broad, immobile butt joints replace the condylar joints between the sternum and sternal ribs. Additionally, the well-developed processes and eminences that were features of the sternum and sternal ribs of *H. tucki* are lost, simplifying the sternal complex in all subsequent clades. Together, this simplification of the sternum decreases the degrees of freedom for associated skeletal components, reducing both the range of motion of the sternocostal complex (relative to the plesiomorphic condition) and its ability to contribute to changes in abdominothoracic volume. The APP is prominent and anteriorly elongated, with anteroposteriorly oriented muscle scars present on the dorsal and medial surfaces (*Morschhauser et al., 2018a*; *Owen, 1875*). In Genasauria, the puboperitoneal muscle is now the major contributor to changes in volume, with the sternum and the abdominal musculature relegated to a secondary role.

Finally, in deeply nested ornithischians (*Figure 8D*), the gastralia remain absent, with the pubic rod shortening to a spur (derived ornithopods) or a tab (derived neoceratopsians), indicating that abdominal musculature now attaches to the ischium and mainly functions to support the viscera. The sternal plates, where present, are relatively small, show no evidence of dorsal rib interaction, and differ markedly in morphology between clades, suggesting that they played no constrained role in ventilation. Convergently, APP area is substantially enlarged and develops clade-specific morphologies. At this stage, the puboperitoneal muscle served as the main ventilatory apparatus, sternal movements contribute little to no volumetric changes, and the non-puboperitoneal abdominal musculature functions mainly as support for viscera.

The changing vertebrocostal orientations along the axial column of *H. tucki* observed here (*Figure 6*) support the bipartite and dorsally immobile lung previously reconstructed in ornithischians and silesaurids (*Schachner et al., 2011*; *Brocklehurst et al., 2018*). Considering that a dorsally immobilized, anatomically and functionally heterogeneous lung has been reconstructed for all of Ornithischia (*Schachner et al., 2011*; *Brocklehurst et al., 2018*), and that the *M. diaphragmaticus* of extant crocodilians is coupled with a flexible lung and shifting viscera, the proposed ventilatory mechanism for Ornithischia was likely functionally distinct from the hepatic piston model present in crocodilians, although the two may have been anatomically convergent. The crocodilian *M. diaphragmaticus* originates on both the pelvis anterior to the acetabulum and the gastralia (or pubic apron, depending upon the taxon) (*Gans and Clark, 1976*). It then fans out anteriorly, encapsulates all of the abdominal viscera (dorsally, laterally, and ventrally), and inserts on the liver, with fibres occasionally extending to the pericardium (*Gans and Clark, 1976*). The proposed puboperitoneal muscle in ornithischians originating on the APP (*Figure 8B–D*) is reconstructed here as travelling anteriorly, and inserting on any potential number of anatomical structures, including the dorsal surface of the liver, pulmonary septa, posteriorly positioned air sacs (or non-invasive pulmonary diverticula emerging from the lung), or even the posterior aspect of the lung itself if no pulmonary diverticula existed. This putative mechanism would be distinctly different from that of extant

crocodilians, particularly in the larger, later-branching ornithischian taxa in that there would be no ventral attachment due to a loss of gastralia and the shortened pubis (*Figure 8D*). Without bundling the abdominal viscera into a fusiform tube, there would be no anterior-posterior translation of the entire visceral mass within the thoracocoelomic cavity. Additionally, ventilation of the anteriorly immobilized respiratory parenchyma by a posterior/ventral flexible region (whether air sacs, or just a flexible sac-like expansion) would not theoretically cause the same shifts in centre of mass that the crocodilian hepatic-piston mechanism does (see, e.g., *Uriona and Farmer, 2008*), and may be more functionally analogous to the complementary integration of pelvic musculature as observed in birds (e.g., *Columba livia*; *Baumel et al., 1990*). This hypothesis posits that only the flexible regions of the lung linked to the pelvic bellows would be stretching with contraction of the muscle, while the anterior and dorsal regions of the lung containing the respiratory parenchyma could remain fixed and immobilized to the adjacent skeletal tissues. This type of pulmonary heterogeneity is well documented in other sauropsids outside of birds (e.g., varanids [*Schachner et al., 2014*], chameleons [*Klaver, 1973*], snakes [*Wallach, 1998*]), where there is an extreme separation of the respiratory parenchyma and more flexible sac-like structures in the posterior region of the lung, and thus supports the possibility of these characters independently evolving in this lineage if this ventilatory mode is truly divergent from other dinosaurs.

## Lung ventilation in dinosaurs is probably more complicated

Investigation into dinosaur respiration has focused on PSP, using its presence or absence as a sole proxy for avian-like ventilation and physiology (*O'Connor and Claessens, 2005*; *O'Connor, 2006*; *Butler et al., 2012*; *Wedel, 2003*). Recent studies showing the remarkable multiplicity of respiratory systems employed by living reptiles (*Owerkowicz et al., 1999*; *Cieri et al., 2018*; *Claessens, 2009*; *Brocklehurst et al., 2017*; *Baumel et al., 1990*; *Farmer and Carrier, 2000*; *Lyson et al., 2014*) show that PSP is only one component of a complex suite of features that coevolve to enable lung ventilation across a swathe of tetrapod lineages. This recent research shows that some presumed 'bird-like' respiratory features, such as unidirectional air flow, are actually plesiomorphies characterizing much larger groups (*Schachner et al., 2014*; *Farmer and Sanders, 2010*) and highlights the diversity of ways in which multiple anatomical systems interlink to effectively ventilate the lungs. New work on the pulmonary anatomy of the ostrich (*Struthio camelus*) has demonstrated that PSP relationships with the respiratory system in extant birds may not be as straightforward as previously thought (*Schachner et al., 2021*), and reconstructions of dinosaur lungs that directly follow a standardized avian *bauplan* may need to be reconsidered. Additionally, primitive features like gastralia, simple sterna, and 'propubic' pelves impede attempts at completely superimposing the highly derived physiology of birds onto comparatively less-specialized clades like non-avian theropods and sauropods.

Inquiry into ornithischian breathing has been stunted by virtue of their phylogenetic position between clades that have received more thorough respiratory evolution investigation (*O'Connor and Claessens, 2005*; *Wedel, 2006*; *Butler et al., 2009*; *Claessens et al., 2009*; *Fechner and Gößling, 2014*; *Zheng et al., 2020*; *Tickle et al., 2012*; *Codd et al., 2008*; *Wedel, 2003*) paired with inferences informed by phylogenetic bracketing (*Witmer, 1995*). As discussed here, ornithischians are outliers among ornithodirans for many reasons – in particular, their unique lung structure and lineage-wide lack of PSP contradict the more parsimonious inferences made about their respiratory anatomy (i.e., that ornithischians are predicted to have conspicuous air sacs).

Archosaurs likely demonstrate a remarkably labile respiratory evolution that has yet to be fully appreciated, and future inquiry is at risk of overlooking a variety of ventilatory mechanisms that are obfuscated by more parsimonious explanations. This suggests that dinosaur, and archosaur, breathing should be investigated with a more nuanced view of evolution; a paradigm that is informed by extant respiratory diversity and that is simultaneously willing to risk relaxing an insistence on phylogenetic bracketing in an attempt to capture increased ventilatory diversity in extinct lineages. Similar trappings will inevitably extend beyond the topic of respiratory evolution, with the evolution and homology of archosaurian integumentary structures being an additional area that will likely struggle from comparable oversimplifications.

The success of ornithischians is remarkable, and the reason for the marked differences between their body plans and those of other dinosaurs remains enigmatic. The diverging ventilatory

adaptations hypothesized here provide an overarching explanation for a wide range of skeletal modifications, and perhaps accompanying metabolic and physiological changes, that shaped the lineage for 130 million years. It is likely no coincidence that the evolution of the APP and its hypothesized role in ventilation precede dramatic and conspicuous increases in ornithischian diversity and disparity.

## Materials and methods

### Statistical analysis

To quantify pelvic evolution in Ornithischia, we measured femoral length, APP (from the anterior margin of the acetabulum), pubic rod (from in line with the anterior margin of the acetabulum to the distalmost tip), and ischial (the contour of the posterior surface that initiates on the iliac peduncle and terminates at the middle of the distalmost point) lengths for a phylogenetically broad sample of ornithischian taxa (*Appendix 1—table 1*) through direct measurements of specimens, high-resolution photos, and published sources. We measured specimens either by hand, using digital callipers and measuring tapes, digitally from 3D SRμCT data, or from high-resolution photos of specimens where scale bars were available and accurate. To normalize scale, we $\log_{10}$-transformed all measurements. We selected pubic and ischial measurements because of their hypothesized relationship with lung ventilation (in particular, plesiomorphic models like cuirassal breathing; *Carrier and Farmer, 2000*). We chose femoral length as a proxy for body mass, even though it has lower correlation coefficients than femoral circumference for body mass estimation (*Campione and Evans, 2012*; *Anderson et al., 1985*). We used it here because circumference measurements were unavailable for most of our specimens and because femoral length is frequently used in the literature and therefore practical to collect (e.g., *Christiansen and Fariña †, 2004*). Although it is unlikely that this choice greatly affects the results we present here, stegosaurs appear to have apomorphically long femora that are likely to affect body mass corrections for these taxa specifically; we accept this localized trade-off in error for the benefit of standardized measurements across our sampling of Ornithischia. We analysed these data using scripts written in the R statistical software language (*R Development Core Team, 2013*) and its associated packages 'ape' (*Paradis et al., 2004*), 'ggplot2' (*Wickham, 2016*), 'phytools' (*Revell, 2012*), 'strap' (*Bell and Lloyd, 2015*), 'geiger' (*Harmon et al., 2008*), and 'nlme' (*Pinheiro et al., 2012*).

To investigate evolutionary patterns in the APP, pubic rod, and ischium, we used pGLS regressions of these pelvic measurements against femoral length and calculated residuals from these regressions. This is a common means of assessing phylogenetic and size-corrected variance in morphological datasets and can be used together with comparative phylogenetic methods (*Revell, 2009*; *Hunt and Carrano, 2010*). We used the residuals as continuous characters to both qualitatively map on the ornithischian tree and to assess the fit of a variety of macroevolutionary models implemented in the R package Geiger (*Harmon et al., 2008*). We used the corrected AICc and computed likelihood ratio tests to assess whether the preferred model is significantly better than the next-best model. '*Source code 1*' is R code to reproduce statistical analysis; '*Source code 2* and *3*' are phylogenetic tree files in .phy and .nex formats, respectively; and '*Source code 4*' is 'First Appearance Date' and 'Last Appearance Date' data of taxa analysed, obtained from the Paleobiology Database (paleobiodb.org).

### Geological context

AM 4766 was recovered from the upper Elliot Formation (uEF) in strata that correlate with the *Massospondylus* Assemblage Zone (*Viglietti et al., 2020*) and is likely Sinemurian in age (*Bordy et al., 2020*). The specimen was recovered from a light red, clast-rich, very fine-grained sandstone that is consistent with palaeo-environmental reconstructions of the uEF as a seasonally wet, fluvio-lacustrine system (*Bordy et al., 2004a*). Further details of the geological context are elaborated in Appendix 1 and figured in *Appendix 1—figures 1* and *2*.

### Digital specimen reconstruction

Volume files of AM 4766 were reconstructed using a combination of manual and semi-automated tools (i.e., pen tool, interpolate function) in Avizo Lite version 9.0 (FEI Visualization Sciences Group,

Merignac, France), with various segmented regions stitched together in VGStudio Max version 3.2 (Volume Graphics, Heidelberg Germany). Detailed synchrotron scanning and data processing protocols are outlined in Appendix 1.

## Conclusion

An exceptionally preserved specimen of the ornithischian dinosaur *H. tucki* reveals novel features of the anatomy of this taxon. Some of these features were previously unknown in Ornithischia, including a complete gastral basket with thin, single-element gastralia; bizarre, paddle-like sternal ribs with prominent condylar articular surfaces; and an apomorphic pair of well-developed sternal plates. Other features present in the specimen have a sporadic distribution in Ornithischia, including an ossified suprascapula and clavicles. These findings support the basal position of *H. tucki* and further support its importance as a transitional taxon showing the early evolution of iconic ornithischian anatomical features. Using SRμCT scans of the specimen, we were also able to observe in *H. tucki* the lack of PSP in the vertebral column, a smooth posterior thoracic ceiling, and a relatively small APP. We conducted a quantitative analysis of relative sizes of pelvic girdle elements and showed that the APP alone evolved in a manner consistent with the predictions of an 'Early-Burst' model, increasing markedly in proportional size early in the diversification of Ornithischia, and then remaining relatively large in all ornithischian lineages, with lower rates of change. These results are explained by a model for the evolution of the ornithischian ventilatory apparatus, in which the lineage undergoes a shift from a hypaxial-dominated system of volumetric change to a system where the lungs are ventilated by a novel pelvic muscle attached to the APP – a muscle functionally analogous to the dorsal component of *M. diaphragmaticus* of extant crocodilians. *H. tucki* preserves evidence for a critical transition in dinosaurs, demonstrating how key innovations evolve, showing how they can have pervasive effects on multiple anatomical systems, and providing a possible explanation for the success and longevity of a major lineage of dinosaurian herbivores.

## Acknowledgements

We acknowledge the European Synchrotron Radiation Facility ESRF for provision of synchrotron radiation facilities and would like to thank Paul Tafforeau for assistance in using beamline ID17. We thank Julien Benoit and Safiyyah Iqbal for assistance with 3D visualization in Avizo; Paul Barrett whose supportive comments and constructive feedback on an earlier version of this manuscript improved it greatly; Roger Benson for consultation on statistical analyses; Bruce S Rubidge and Wilma Wagener for administrative support and encouragement; Leon Claessens and Corwin Sullivan for interesting discussions about breathing; Clint Boyd and Cathy Forster for supplying photographs of important taxa; John Hepple for assisting in excavating AM 4766; Ben Maclennan for alerting WJDK to the fossil locality; the Siberia farm owner, Johannes van Heerden; and Zaituna Skosan and Rose Prevec of the Iziko South African and Albany Museums, respectively, for facilitating access to specimens. We also thank Genya Masukawa for granting us permission to modify his *Thescelosaurus* skeletal figure for use in *Figures 4* and *8*.

## Additional information

### Funding

| Funder | Grant reference number | Author |
| --- | --- | --- |
| DST-NRF Centre of Excellence in Palaeosciences | | Viktor J Radermacher |
| Palaeontological Scientific Trust | | Viktor J Radermacher |
| Durand Foundation for Evolutionary Biology and Phycology | No. DFEBP00001/16 | Viktor J Radermacher |
| DST-NRF-African Origins Platform | 98800 | Jonah N Choiniere |
| DST-NRF-African Origins Platform | 98825 | Emese M Bordy |

The funders had no role in study design, data collection and interpretation, or the decision to submit the work for publication.

## Author contributions
Viktor J Radermacher, Conceptualization, Formal analysis, Investigation, Visualization, Methodology, Writing - original draft; Vincent Fernandez, Data curation, Software, Formal analysis, Investigation, Visualization, Methodology, Writing - original draft, Project administration; Emma R Schachner, Conceptualization, Validation, Investigation, Writing - original draft; Richard J Butler, Validation, Investigation, Writing - original draft; Emese M Bordy, Investigation, Writing - original draft; Michael Naylor Hudgins, Resources, Writing - review and editing; William J de Klerk, Data curation, Supervision, Funding acquisition, Investigation, Writing - review and editing; Kimberley EJ Chapelle, Supervision, Project administration, Writing - review and editing; Jonah N Choiniere, Conceptualization, Formal analysis, Supervision, Funding acquisition, Investigation, Methodology, Writing - original draft, Project administration

## Author ORCIDs
Viktor J Radermacher https://orcid.org/0000-0001-6524-7811
Vincent Fernandez http://orcid.org/0000-0002-8315-1458
Emma R Schachner https://orcid.org/0000-0002-8636-925X
Richard J Butler https://orcid.org/0000-0003-2136-7541
Emese M Bordy https://orcid.org/0000-0003-4699-0823
Michael Naylor Hudgins https://orcid.org/0000-0003-0215-7792
Kimberley EJ Chapelle https://orcid.org/0000-0002-9991-0439
Jonah N Choiniere https://orcid.org/0000-0002-1008-0687

## Decision letter and Author response
Decision letter https://doi.org/10.7554/eLife.66036.sa1
Author response https://doi.org/10.7554/eLife.66036.sa2

# Additional files
## Supplementary files
- Source code 1. CSV file of taxa and measurements used in analysis.
- Source code 2. Phylogenetic tree file in Simple Newick format.
- Source code 3. Phylogenetic tree file in Nexus format.
- Source code 4. R script used in analysis.
- Source code 5. CSV file of First Occurrence Dates (FAD) and Last Occurrence Dates (LAD) of taxa used in analysis.
- Transparent reporting form

## Data availability
All data generated or analysed during this study are included in the manuscript and supporting files in .csv and .R formats.

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

## Appendix 1

### Synchrotron characterization protocol

Specimen AM 4766 was imaged on the ID17 beamline of The European Synchrotron (ESRF, Grenoble, France). The experiment consisted of two parts: (1) imaging the whole of each block and (2) re-imaging specific regions of interest at higher resolution (i.e., the sternal region where anatomy is obscured by matrix and larger parts of the skeleton). The beamline was setup for propagation phase-contrast synchrotron X-ray micro-computed tomography: a monochromatic beam of 160 keV (Laue double bent Si 111), a sample-detector distance of 11 m and an indirect detector comprising a 2 mm LuAG scintillator, a set of lenses with a magnification of ~0.3×, and a visible light detector (CCD or sCMOS). For the first set of experiments, we used a FReLoN 2k camera (ESRF), recording 4998 radiographs over a 360° rotation of the sample with a pixel size of 46.95 μm. Each radiograph consisted of five accumulated images, each with an exposure time of 40 ms, resulting in a total exposure time of 0.2 s per recorded radiograph. For the second part of the experiment, we used a PCO. edge 5.5 sCMOS with camera link (PCO, Kelheim, Germany), recorded 6000 projections with a pixel size of 21.80 μm. The total exposure time was 0.1 s (four accumulated images of 25 ms). For both sets of experiments, the centre of rotation was shifted near the edge of the field of view, almost doubling the size of reconstructed slices, keeping a margin ranging from 10% to 20% to ensure an overlap between projections 180° apart.

Given the limited vertical size of the X-ray beam (ca. 7.5 mm), the full acquisition of a sample required several scans, moving various blocks of the specimen by 50% of the vertical field of view between each scan. The significant overlap was used to stitch radiographs vertically before tomographic reconstruction. As the PCO.edge 5.5 produces a vertically banding noise, each image was de-noised: first, a 1D high-pass median filter with a width of 5 pixels was applied horizontally; second, a three passes 1D vertical median filter with a width corresponding to 50% of the vertical size of the image. The overlapping part of two consecutive scans required a registration: the horizontal alignment was based on the position of the centre of rotation, while the vertical alignment was evaluated based on the mean error around the predicted value. Computation of the overlapping part was handled in two parts. The low frequencies (based on a low-pass median filter of a few pixels in width) were obtained using a weighted average, based on the vertical intensity profile of the beam. High frequencies (HFs) were handled to diminish the presence of strong ring artefacts in reconstructed slices: first, a blurred averaged of both HF images was generated; then a final image was generated, selecting pixels from both HF images that were less divergent from the blurred averaged image.

Tomographic reconstruction was done using PyHST2 (*Mirone et al., 2014*) using the single distance phase retrieval approach (*Paganin et al., 2002*). The δ/β parameter (indicating the nature of the material for single-phase material) was set to reduce the noise while avoiding blurring of dense metallic inclusions. To reduce the impact of dense inclusions in the data, we performed a normalization of artifactual grey-level gradients and metallic inclusion correction (*Cau et al., 2017*). Remaining processing included modification of the bit depth from 32 to 16 bits, using the 0.001% minimum and maximum exclusion values of the 3D histogram generated by PyHST2; ring correction (*Lyckegaard et al., 2011*); and cropping the volume. Finally, a 2 × 2 × 2 binning was applied to increase the signal-noise-ratio and reduce the size of the data for segmentation.

### Phylogenetic tree used

Ornithischian systematics have recently received renewed attention, with multiple new interesting but contentious relationships hypothesized (*Müller and Garcia, 2020*; *Dieudonné et al., 2020*; *Yang et al., 2020*; *Baron et al., 2017a*). Much of this recent inquiry is worthy of attention and further interrogation, but support for these novel interrelationships of major ornithischian clades remains low. Because of this, we opted to remain conservative and use a more traditional version of ornithischian relationships that has consistently been recovered in multiple analyses (*Butler et al., 2008*; *Boyd, 2015*; *Baron et al., 2017b*) with Heterodontosauridae repeatedly recovered as the basalmost clade of Ornithischia. The phylogenetic position of *Eocursor parvus* largely remains poorly resolved and weakly supported (*Barrett et al., 1791*; *Rozadilla et al., 2019*; *Cruzado-Caballero et al., 2019*; *Andrzejewski et al., 2019*; *Madzia et al., 2018*). We ultimately favoured a

phylogenetic position for this taxon derived from analyses that inspected the holotype specimen (SAM-PK-K8025) first-hand (*Baron et al., 2017a*; *Baron et al., 2017b*) and recovered it as a basal neornithischian. Ornithopodan relationships were compiled from multiple sources (*Rozadilla et al., 2019*; *Cruzado-Caballero et al., 2019*; *Prieto-Marquez et al., 2016*) to encompass both basal and derived taxa, with marginocephalians relationships derived from a comprehensive appraisal of ceratopsian relationships (*Morschhauser et al., 2018b*).

Changes in the phylogenetic position of heterodontosaurids and *E. parvus* are likely to affect our inferences here. However, studies that posit a late-branching topology for these taxa remain tenuous and significantly temporally incongruent (*Dieudonné et al., 2020*).

## Geological context of AM 4766

### Stratigraphic provenance and age

AM 4766 was found in the uEF (Stormberg Group, Karoo Supergroup) on Farm Alpha 40, 0.8 km southwest of the Siberia farmhouse in the Franshoek se Loop stream bed in the Eastern Cape Province of South Africa (S31°7′42′; E27°17′38′; *Appendix 1—figure 1*). In addition to exposures of the uEF, the study area also exposes the conformably overlying, mainly aeolian Clarens Formation and succeeding Karoo volcanics (*Appendix 1—figure 1A–C*). The latter forms part of the late Pliensbachian to early Toarcian Drakensberg Group, which in the study area is dominated by continental flood basalts and interbedded sandstone layers (*Duncan et al., 1997*). Based on the biostratigraphic subdivision of the Elliot Formation (e.g., *Kitching and Raath, 1984*; *McPhee et al., 2017*), the rocks at the study site are within the upper *Massospondylus* Assemblage Zone (*Viglietti et al., 2020*) and are likely Sinemurian (*Bordy et al., 2020*). Less than 500 m to the NE from the fossil locality, a remarkable syn-sedimentary normal fault displaces the uppermost Elliot and Clarens Formations (see Figure 9a in *Bordy, 2004b*). Immediately south of the fossil locality, a deeply incised, palaeovalley cuts across the Clarens and into Elliot Formations and is filled with lava flows and sandstone beds of the Drakensberg Group (e.g., *Du Toit, 1906*).

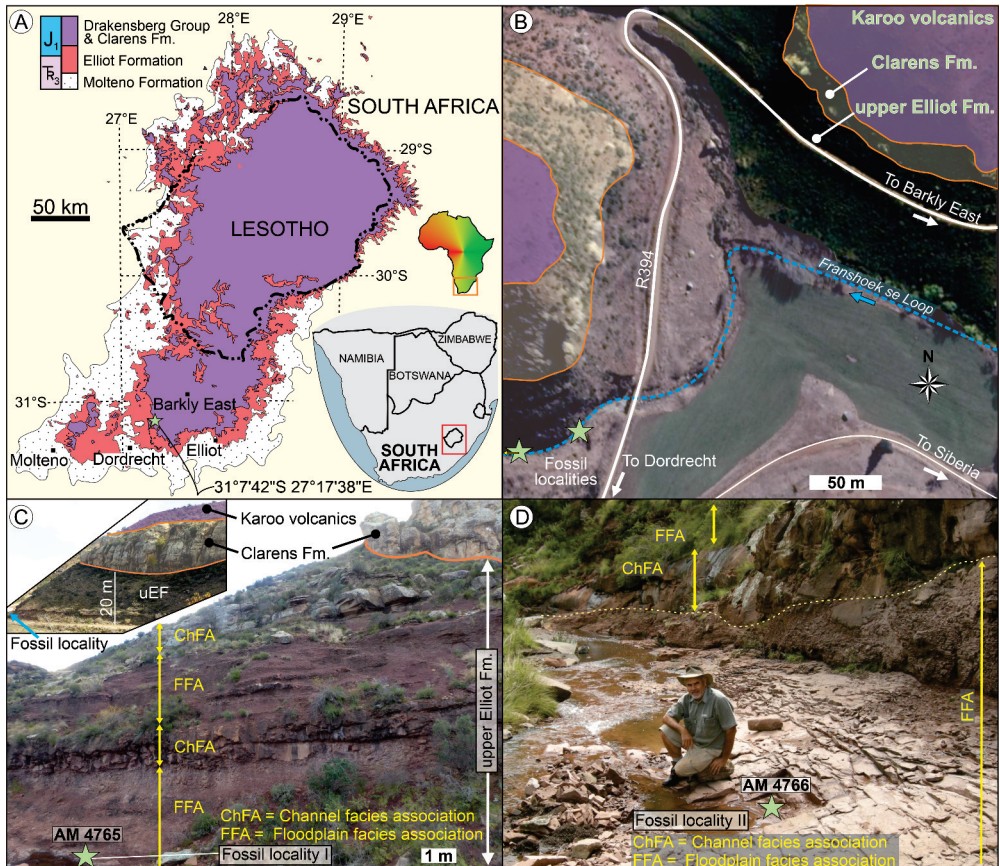

**Appendix 1—figure 1.** Stratigraphic context and locality maps of the fossil specimens. (**A**) Simplified geological map of the upper Karoo Supergroup in the central main Karoo Basin. Star marks the exact study locality in the Eastern Cape of South Africa. Map derived from Council for Geoscience (2008). (**B**) Simplified geological map of the study area west of the Siberia farmhouse. Fossil localities are <90 m apart in the streambed. Map derived from combining data of the Council for Geoscience (2008), Google Earth 2010, and own mapping. (**C**) Outcrop near the locality of AM 4765 (view from S), which is ~25 m below the contact of the upper Elliot and Clarens Formations. Inset shows the uneven basal contact of the Clarens Formation in the region (view from the NE; uEF: upper Elliot Formation). (**D**) Outcrop of upper Elliot Formation near the locality of AM 4766 (view from East), which is <90 m downstream from AM 4765.

## Sedimentology

### Description

At the study locality, the fluvial redbed succession of the uEF comprises two deep red, maroon to deep pink facies associations (*Bordy et al., 2004a*; *Bordy, 2004b*): the Channel and Floodplain Facies Associations (ChFA and FFA, respectively). The ChFA (*Appendix 1—figure 1C, D*) is sandstone-dominated and contains multi-storey sheet-like, medium- to thickly bedded sandstone strata with sharp, even external and internal bounding surfaces that can be followed laterally for several hundred metres. The fine- to very fine-grained feldspathic wackes within the ChFA are either massive or show horizontal lamination, ripple cross-lamination, and rare planar cross-stratification. Thin, localized conglomerate layers/lenses of intraformational mud-pebble clasts and reworked carbonate nodules are common. The FFA (*Appendix 1—figures 1C, D,* and *2*) is siltstone-dominated and contains (1) clast-rich, massive, silty, very fine-grained sandstone (facies Sc); (2) massive silty, very fine-grained sandstones with erosive lower and gradational upper contacts (facies Sm); (3) massive siltstone beds (facies Fm); (4) horizontally laminated very fine-grained sandstones (facies Sh); and (5) low-angle cross-bedded sandstone (facies Sl; *Appendix 1—figure 2A–D*). All facies in the FFA are

pedogenically altered, and thus desiccation cracks, in situ pedogenic carbonate nodules, calcretized and clay-lined root traces, and invertebrate trace fossils are very common (*Appendix 1—figure 2*). The light red, massive, clast-rich, very fine-grained sandstone, a diagnostic sedimentary facies of the uEF (see, e.g., pages 393, 395, 397 of *Bordy et al., 2004a*; and pages 366, 369 of *Bordy et al., 2017*), contains the *Heterodontosaurus* fossils (AM 4765, AM 4766) at the study site. Facies Sc is readily recognizable because of its matrix-supported fabric comprising silty, fine- to very fine-grained sand as matrix and intraformational rip-up clasts of mudstone and sandstone (*Appendix 1—figure 2*). The clasts are angular to very angular and poorly sorted, ranging in diameter from granules to large pebbles (mostly <5 cm, rarely ~20 cm in diameter – see *Appendix 1—figure 2D*). The sandstone clasts in facies Sc at this site are typically very fine-grained, massive or laminated or ripple cross-laminated sandstone (*Appendix 1—figure 2C, D*). Lateral dimensions of facies Sc are not fully exposed in this outcrop; however, its sharp, erosional contacts with the interbedding facies Sm, Sl, and Sh and overlying facies of ChFA are well-exposed (*Appendix 1—figure 1C, D*, *Appendix 1—figure 2A–E*).

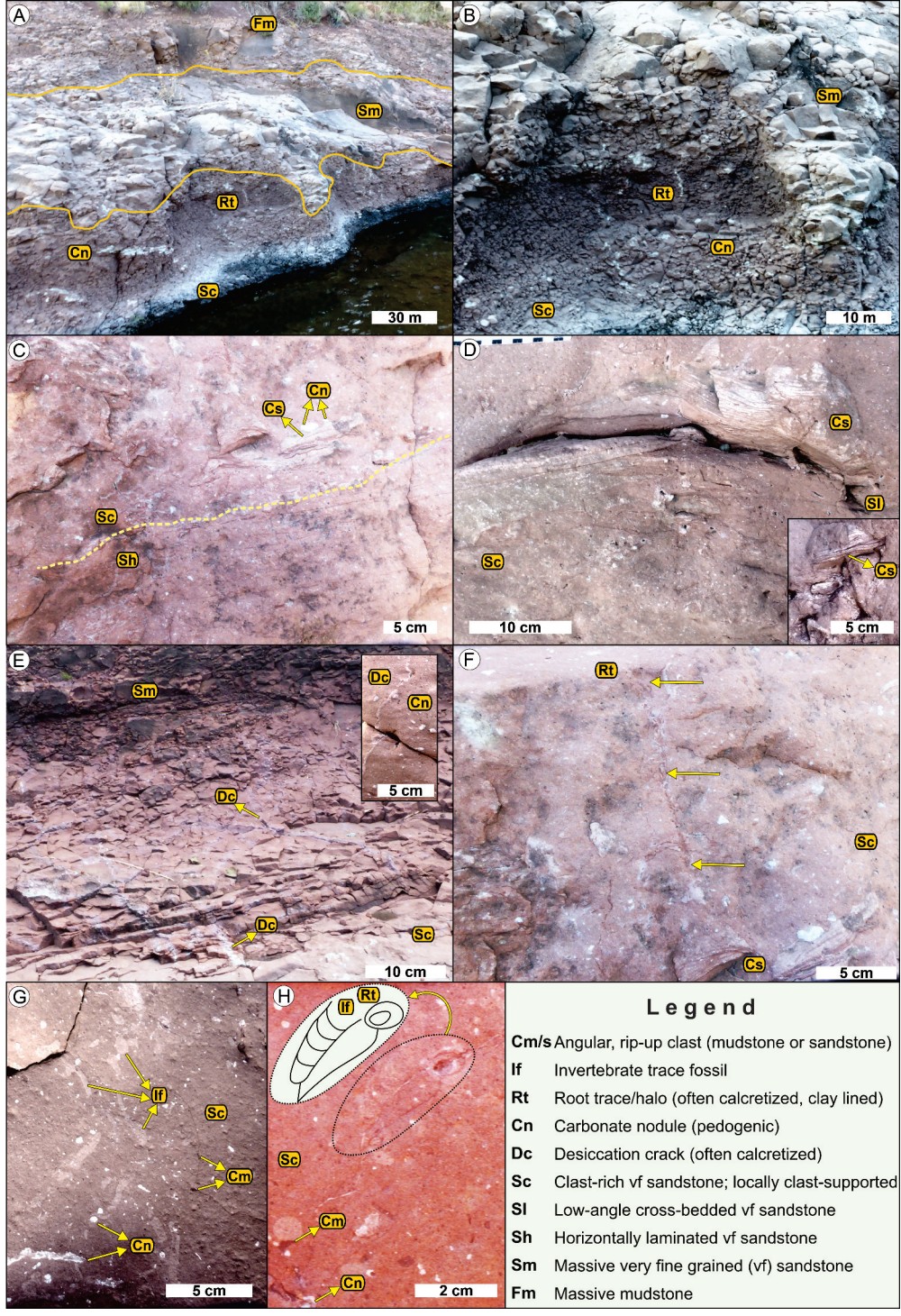

**Appendix 1—figure 2.** Sedimentological context of the fossil specimens in the upper Elliot Formation. See legend for facies codes. (**A**) Outcrop view of the floodplain facies association. From base to top: clast-rich, massive, pedogenically altered (Cn, Rt) silty, very fine-grained sandstone bed (Sc); massive silty, very fine-grained sandstone bed (Sm) with erosive lower and gradational upper contacts; massive siltstone bed (Fm). (**B**) Close-up view of the erosive contact between facies Sc and Sm. (**C**) Horizontally laminated sandstone (Sh) overlain by clast-rich, massive silty, very fine-grained sandstone with rip-up clasts of laminated sandstone (Cs) and in situ pedogenic carbonate nodules (Cn). (**D**) Low-angle cross-bedded sandstone (Sl) under- and overlain by clast-rich, massive silty, very

*Appendix 1—figure 2 continued on next page*

*Appendix 1—figure 2 continued*

fine-grained sandstone. The upper Sc facies contains large, very angular rip-up clasts (see inset too) of ripple cross-laminated sandstone (Cs). (**E**) Outcrop view of the polygonal network of desiccation cracks (Dc) in facies Sc. Inset shows the close-up view of the calcretized desiccation cracks (Dc) and in situ pedogenic carbonate nodules (Cn). (**F**) Vertical root trace with mineral halo, clay lining, and pedogenic alteration in facies Sc. Preserved length of root (~15 cm) is marked by yellow arrows. (**G**) Network of invertebrate trace fossil (If), poorly sorted rip-up clasts (Cm), and in situ pedogenic carbonate nodules (Cn) in facies Sc. (**H**) Back-filled burrow cast of an invertebrate trace fossil (If) adjacent to a clay-lined root trace (Rt) in facies Sc. Note the poorly sorted rip-up clasts (Cm) and in situ pedogenic carbonate nodules (Cn). Inset shows the line drawing of the trace fossils.

## Interpretation

The broad, laterally continuous tabular sheets of very fine to fine-grained sandstone with erosional contacts and upper flow regime sedimentary structures (facies Sh, Sm) in the ChFA in the uEF are interpreted as shallow but broad channel fills and sheet wash sediments that formed during high-energy, short-lived flash flood events (see, e.g., *Bordy et al., 2004a*; *Bordy, 2004b*; *Bordy et al., 2017* and references therein). The rest of the facies making up the FFA in the uEF are considered products of lower-energy, lower-gradient settings where short-duration, but high-energy depositional events (stream flows, debris flows, hyperconcentrated flows) were restricted laterally to localized depressions. Specifically, the fossil-bearing clast-rich sandstone beds (facies Sc) are interpreted as products of hyperconcentrated flows, which are intermediate between normal stream and debris flows (e.g., *Lowe, 1988*; *Benvenuti and Martini, 2002*). These sudden events were able to remove and transport *en masse* large amounts of silty, sandy sediments, including semi-consolidated, intra-formational clasts. Due to the large volume of rapidly mobilized debris, these short-lived sediment flows could also entomb animals both alive and dead. These flows filled the smaller, rainstorm-eroded gullies and other irregular depressions of the floodplain area, and in turn were later eroded by small streams that crisscrossed the depositional surface of the floodplain. All facies of the FFA have experienced extensive desiccation and pedogenic overprinting, which included colonization by plants and burrowing invertebrates (*Appendix 1—figure 2E–H*). This reconstruction is in line with palaeo-environmental interpretation of the uEF (e.g., *Bordy et al., 2004a*) as a continental redbed succession that was deposited in a flash flood-dominated, seasonally wet and increasingly arid fluvio-lacustrine system in southern Africa during the Sinemurian.

**Appendix 1—table 1.** List of taxa used in the study.

Asterisk (*) indicates measurements derived from mostly complete specimens scaled relative to each other in a shematic skeletal diagram

| Taxon | Specimen no. | Source and notes |
|---|---|---|
| *Heterodontosaurus tucki* | AM 4766; SAM-PK-K1332 | First-hand, Synchrotron X-ray µCT data |
| *Eocursor parvus* | SAM-PK-K8025 | *Butler et al., 2007* |
| *Scelidosaurus harrisonii* | NHMUK R1111 | *Norman, 2020* |
| *Kentrosaurus aethiopicus* | MFN mount | *Mallison, 2010* |
| *Stegosaurus stenops* | NHMUK PV R36730 | *Maidment et al., 2015* |
| *Agilisaurus louderbacki* | ZDM 6011 | Photographs |
| *Hexinlusaurus multidens* | ZDM T6001 | *Xl and Cai, 1984* |
| *Jeholosaurus shangyuanensis* | IVPP 15719 | Photographs, first-hand, **pubic rod length estimated relative to ischial length and phylogenetic position |
| *Haya griva* | IGM 100/2015 | Photographs, *Makovicky et al., 2011* |
| *Dryosaurus altus* | YPM 1876 | *Marsh, 1878* |
| *Nanosaurus consors* | BYU ESM-163R | Photographs |
| *Thescelosaurus neglectus* | NCSM 15728 | Photographs, first-hand |
| *Psittacosaurus neimongoliensis* | IVPP 12-0888-2 | *Russell and Zhao, 1996* |
| *Centrosaurus apertus* | AMNH 5351 | *Brown, 1917* |
| *Styracosaurus albertensis* | AMNH 5372 | *Brown and Schlaikjer, 1937* |
| *Utahceratops gettyi** | UMNH VP specimens | *Sampson et al., 2010*, |
| *Homalocephale calathocercos* | GI-SPS 100/51 | *Maryanska and Pachycephalosauria, 1974* |
| *Tenontosaurus tilletti* | AMNH 3040 | *Forster, 1990* |
| *Iguanodon bernissartensis* | IRSNB 1534 | *Paul, 2008* |
| *Parasaurolophus cyrcocristatus* | FMNH P27393 | *Ostrom, 1963* |
| *Lambeosaurus magnicristatus* | TMP 66.04.01 | *Evans and Reisz, 2007* |
| *Corythosaurus casuarius* | AMNH 5240 | *Brown, 1916* |
| *Brachylophosaurus canadensis* | MOR 794 | *Prieto-Marquez, 2001* |
| *Edmontosaurus regalis* | CMN 2289 | *Campione, 2015* |

Institutional abbreviations: AM: Albany Museum, Makanda, Eastern Cape, South Africa; AMNH: American Museum of Natural History, New York, United States; BYU: Brigham Young University Earth Science Museum; CMN: Canadian Museum of Nature, Ottawa, Ontario; FMNH: Field Museum of Natural History, Chicago, Illinois; GI-SPS: Geological Institute Section of Paleontology and Stratigraphy, Mongolian Academy of Sciences, Ulaanbataar, Mongolia; IGM: Institute of Paleontology and Geology, Mongolian Academy of Sciences, Ulaanbaatar, Mongolia; IRSNB: Institut Royal de Science Naturelle de Belgique, Brussels, Belgium; IVPP: Institute of Vertebrate Paleontology and Paleoanthropology, Beijing, China; MFN: Museum für Naturkunde, Berlin, Germany; MOR: Museum of the Rockies, Bozeman, Montana; NHMUK: Natural History Museum, London, England; NCSM: North Carolina Museum of Natural Sciences, Raleigh, North Carolina; SAM: Iziko South African Museum, Cape Town, South Africa; TMP: Royal Tyrrell Museum of Paleontology, Drumheller, Canada; UMNH: Utah Museum of Natural History, Salt Lake City, Utah; YPM: Yale Peabody Museum, New Haven, Connecticut; ZDM: Zigong Dinosaur Museum, Sichuan, China.

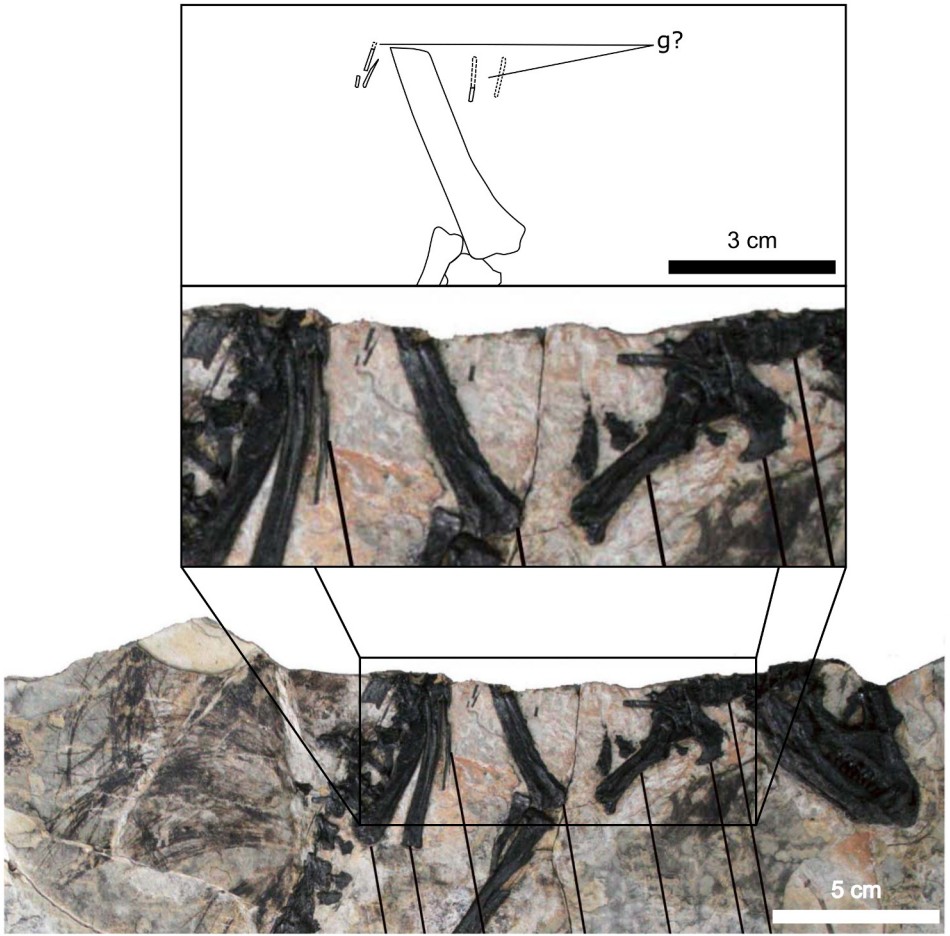

**Appendix 1—figure 3.** Possible gastralia in the *Tianyulong* holotype. Inset shows close-up of tiny rod-like bones that bear some resemblance to the posteriorly diminished gastralia of AM 4766. g?: possible gastralia (fragments). Modified from *Zheng et al., 2009*.

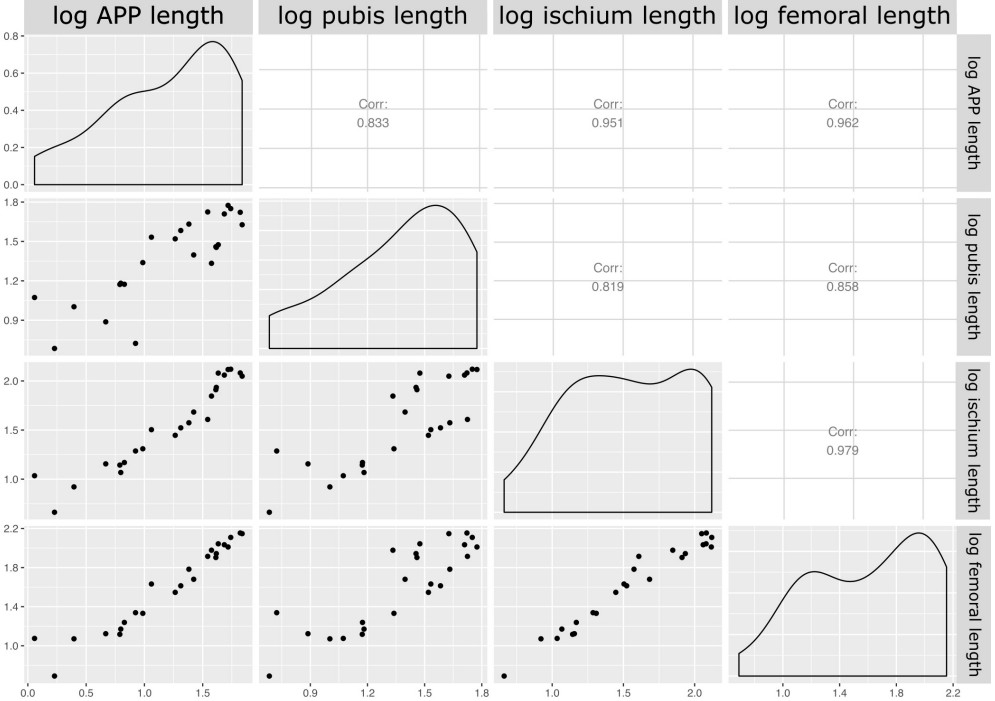

**Appendix 1—figure 4.** Matrix of correlations between measured variables for quantitative analysis of pelvic architecture. Corr: correlation coefficient for relationship; logaleng: log APP length; logpubl: log pubic rod length; logischl: log ischial length; logfeml: log femoral length.

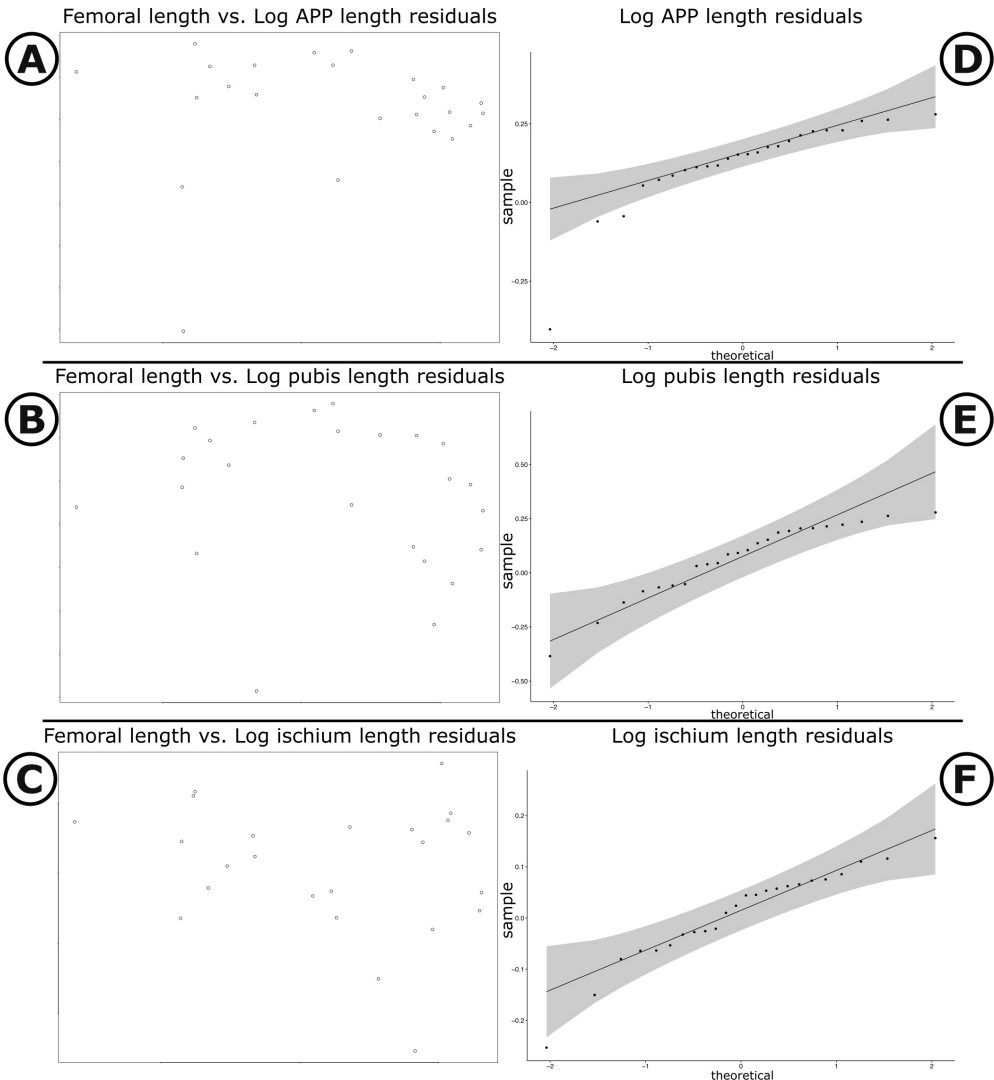

**Appendix 1—figure 5.** Scatter plots and Q-Q plots of pubic measurements versus femoral length. (**A–C**) Scatter plots of log femoral length versus log APP, log pubic rod, and log ischium length, respectively. (**D–F**) Q–Q plots of residuals for log APP, log pubic rod, and log ischium, respectively. Grey shading illustrates 95% confidence intervals.

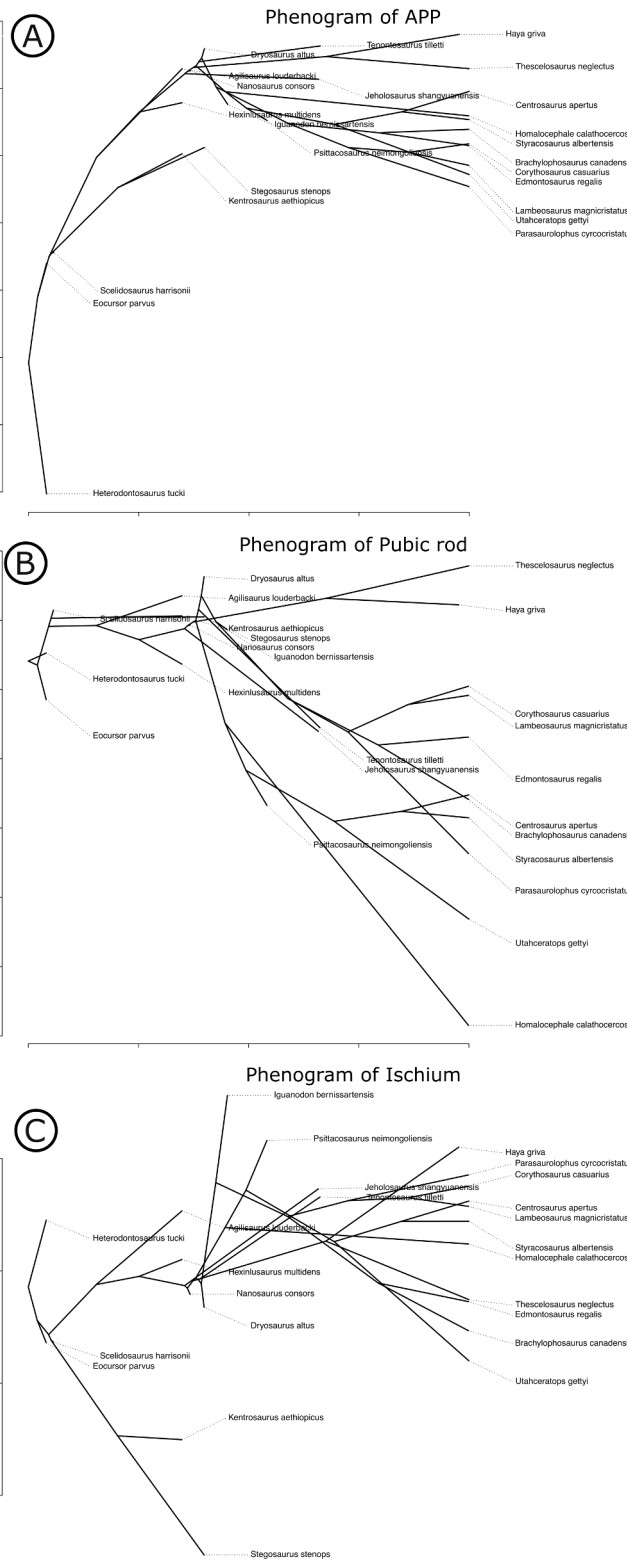

**Appendix 1—figure 6.** Phenograms of measured pelvic dimensions across Ornithischia. (**A**) Anterior pubic process length, (**B**) length of pubic rod, and (**C**) length of the ischium.

