## [Decision Letter]

**Acceptance summary:**

This paper presents important new ideas about the evolution of the ventilatory system and how it is relevant to ornithischian evolution, and presents significant new anatomical information on the basal ornithischian dinosaur *Heterodontosaurus*. It will be a well-received contribution to our understanding of dinosaur physiology and anatomy and contribute to future phylogenetic discussions.

**Decision letter after peer review:**

Thank you for submitting your article "A new *Heterodontosaurus* specimen elucidates the unique ventilatory macroevolution of ornithischian dinosaurs" for consideration by *eLife*. Your article has been reviewed by 3 peer reviewers, one of whom is a member of our Board of Reviewing Editors, and the evaluation has been overseen by George Perry as the Senior Editor. The following individual involved in review of your submission has agreed to reveal their identity: Marc Spencer (Reviewer #3).

This paper provides important new anatomical findings in the basal ornithischian dinosaur *Heterodontosaurus* with conclusions presenting a novel respiratory system in this clade of dinosaurs. The CT scanning of the specimens allowed detailed description of the new skeletal structures which were key to determining the functional significance of the thoracic region.

The reviewers have discussed their reviews with one another, and the Reviewing Editor has drafted this to help you prepare a revised submission. All the reviewers found the manuscript very well-written and illustrated, offering only minor corrections and suggestions to attend to (please see the below reviews), before submitting the final manuscript. Please take note of the points below and we hope you will continue to support *eLife*.

*Reviewer #1:*

This is a well-researched paper describing new anatomical features of well preserved new *Heterodontosaurus* skeleton, that has been imaged using CT scanning to reveal internal anatomy of the skeleton. The descriptions are clear and well-written and the paper is well illustrated, and brings in novel interpretations about respiratory evolution in orthischian dinosaurs. the authors have achieved their aims at describing skeletal structures such as gastralia., sternal plates and ribs, and the internal thoracic ceiling all relevant to interpretation of respiration in ornithischian dinosaurs. The gastral basket is the first described for an ornithischian dinosaur. The dorsoventrally expanded dorsal and ventral ribs were attachment sites for intercostal muscles. the paper presents strong arguments that changes in the anterior pubic process played more of a role in ventilation rather than locomotion for this species, and they describe a unique method of ventilation for ornithischians called 'pelvic bellows".

Overall I find very little to criticise in this paper, it is well argued, with strong statistical support of the changes in key bone structures, good anatomical information about novel structures identified in the new specimen, and good discussion about respiratory evolution in basal ornithischians and how it might have changed during the emergence of the higher clades.While conclusions about the novel breathing in ornithischians might be still a little speculative, the arguments are put forwards in a convincing way, so the hypothesis remains to be further tested with other anatomical explorations of different basal ornithischian taxa. I do not have any suggestions to improve the manuscript, but suggest a thorough check for tightening up text in places.

*Reviewer #2:*

The authors report on a newly-discovered specimen of the early ornithischian dinosaur *Heterodontosaurus*, a taxon exhibiting divergent morphology (compared to later members of the group) and key to many recent cladistic analyses. Synchrotron scanning of this exquisite specimen reveals the presence of gastralia (belly ribs), the first unambiguous occurrence of this feature in Ornithischia, as well as other unusual characteristics such as bizarre sternal ribs, sternal plates, and ossified clavicles and suprascapular. By comparing these and other anatomical features (vertebral and pelvic morphology) across a broad range of ornithischians using quantitative statistical methods, as well as considering data from extant animals especially crocodilians and birds, the authors hypothesize a unique ventilation system in ornithischians – the pelvic bellows – and the key role of *Heterodontosaurus* as a transitional animal preserving the early steps in the evolution of this mechanism.

Strengths: The conclusions of this paper are well-supported by multiple lines of data. Careful anatomical observations and phylogenetically-corrected analyses demonstrate the evolution of key pelvic elements in ornithischians, leading to new functional interpretations of these features and shedding light on the evolution of breathing in dinosaurs. For example, the authors disprove the previous assumption that retroversion of the pubis in early ornithischians required simultaneous decoupling and loss of the gastralia. They also demonstrate the presence of sternal ribs and plates in early ornithischians, but also the fact that these elements more closely resemble those of pterosaurs and theropods than the divergent morphology seen in later ornithischians. Along with increased size of the APP (and reduction in the pubic rod as well as eventual loss of the gastralia) the authors hypothesize the appearance of a novel ventilation system – the pelvic bellows – in ornithischians. The appearance of this mode of breathing occurred just prior to the major radiation of ornithischians within the Jurassic, and the authors trace its step-wise evolution in later ornithischian groups. The authors convincingly rule out locomotor changes as driving changes in APP morphology, and the paper nicely compares similarities and differences between the novel proposed ornithischian ventilation mode and ventilatory systems in extant crocodilians, birds, and lizards. I also agree with the authors' suggestion (near the end of the Discussion) that future studies consider the diversity of ventilatory (and other) systems in extant taxa and therefore the possibility of more diversity in fossil forms.

In sum, the authors achieve their initial aims within this paper and their conclusions are well-justified by the data. This work advances our knowledge of ornithischian anatomy and how archosaurs breathed, and the data provided will be invaluable for future work in dinosaur anatomy, taxonomy, cladistics and ecology, particularly the synchrotron scan data.

Weaknesses: This paper has no technical or conceptual issues – as noted above, the methods are sound and well-applied, and interpretations are well-supported by the data. The paper also benefits from beautiful, very helpful figures.

The only concern regards availability of the imaging (synchrotron) data. There is no mention anywhere in the text about how this can be obtained for reproducibility and future work.

No additional analyses/data are needed and both the science and its presentation are strong, there is no need for improvement in these areas.

My only moderate concern is – there is no reference in the text that I can find on how to access arguably the most critical data within this manuscript, namely the synchrotron data of AM 4766. Either the 3D models (as STLs or 3D PDFs) or the original scan data must be made available with publication. Stating that "data are available upon request" really is no longer acceptable in such studies so dependent on medical imaging and 3D visualization.

Figure 2: For (A) and (C) (specimen photographs), it would be helpful to denote the sternal plates. Also, there should be hyphens for SAM-PK-1332. In the description for (D), it should read SAM-PK-1332, not SAM-PK-1336.

Figure 4: The lettering in the description is mislabeled. It should be (A, B), (C, D), (E), (F), and (G).

Figures 3, 4, and 6: These figures could do well with a specimen photo for reference.

Figure 7: This is described well in the text but it feels like the associated phenograms are a bit too busy and almost detract from the salient points of the figure. Consider revising or explaining the phenograms better in the text. Or perhaps move the phenogram portion to the supplemental information section.

I appreciate Figure 8 in the discussion. You describe the comparisons to other taxa (e.g., crocodylians, avians, non-avian theropods) well in the text. I think an additional figure similar to Figure 8 illustrating these other taxa would serve your discussion very well.

*Reviewer #3:*

Radermacher et al. set out to evaluate the under-studied ventilatory mechanism for ornithischian dinosaurs with the description of a new specimen of the *Heterodontosaurus* (AM 4766) unearthed from the Lower Jurassic upper Elliot Formation of South Africa. It has been postulated in the literature that a protoavian ventilatory mechanism is potentially plesiomorphic for Ornithodira, particularly with immobile non-compliant lungs and air sacs like extant birds. Yet, ornithischian morphology does not follow suit, lacking any known air sacs in the vertebral column or elsewhere (postcranial pneumaticity) as well as lacking osteological evidence for other breathing apparatuses such as gastralia. This new specimen of *Heterodontosaurus*, however, potentially changes these assumptions.

Using synchrotron radiation µCT (SRµCT), the authors document that AM 4766 preserves gastralia (unique among ornithischians), well-developed sternal plates, and potentially mobile sternal ribs. These latter two are known from only a few other ornithischian taxa (e.g., Thescelosaurus, Nanosaurus). The gastralia include a complete series that follows the ventral midline from the sternal plates to the distal pubic shaft, and gastral baskets have been noted to be associated with ventilatory mechanisms in a variety of taxa.

Through a survey of ornithischian taxa that preserve pelvic elements, the authors suggest that a decoupling of the gastral basket of basal ornithischians is associated with the development of the anterior pubic process (APP) that becomes very well developed in later ornithischians. This leads the authors to suggest a new model of ventilation they term the "pelvic bellows" associated with a partially compliant posterior lung and a proposed muscle they call the "puboperitoneal muscle," analogous to the crocodylian M. diaphragmaticus and its hepatic piston model.

Strengths:

The description of the new specimen, AM 4766, is detailed very well. The authors and interested readers know that *Heterodontosaurus* postcranial anatomy has been described and documented several times beginning with Santa Luca's (1980) description of SAM-PK-1332, so they focused on what is new and revelatory about AM 4766.

Radermacher et al. do a convincing job of detailing the presence of gastralia, sternal ribs, and sternal plates and their roles in ventilation among at least *Heterodontosaurus*. Their tentative identification of the few thin fragments of bone preserved in the holotype of Tianyulong confuciusi is also compelling. Though not present in AM 4766, the sternal plates of SAM-PK-1332 preserve a nob-like projection the authors term a costal process, which would articulate with the sternal ribs. The sternal ribs, in turn, possess an articulation with the distal dorsal ribs similar to what is noted in pterosaurs and crocodylians.

The results from the statistical tests (e.g., phylogenetic generalized least squares regressions) do a nice job to help reinforce the connection of the modification of the pelvic architecture, including the APP, with the changing ventilatory mechanism.

The figures resulting from the SRµCT-rendered data are quite illuminating and give the authors' descriptions (and assumptions) some needed weight. This is especially true in the supplementary information section where they outline the geological context of the specimen locality.

Weaknesses:

While I agree with the authors' overall conclusions, it must be stated that this is still relatively speculative. The majority of the paper is describing in detail the morphology of AM 4766 and how that particular specimen (and, presumably, all *Heterodontosaurus*) would ventilate. The Discussion section, which is substantial, largely focuses on the evolution of the pelvic bellows. And, given the statistical tests they ran, it is a reasonable argument; however, I think more could be done with the description of the hypothetical mechanism as one moves through the ornithischian tree into other clades.

Conclusion:

Overall, this manuscript is a very welcome addition to the literature of varying ventilatory mechanisms in archosaurs. It is sorely needed for ornithischians as very little work has been done regarding their breathing apparatuses. More work needs to be done to investigate the appropriateness of this model in more derived clades but this paper is a good starting point for future analyses.

There is not much needed for improvement of this manuscript. I would add a bit more to the comparison between other taxa and how analyses and model development for their ventilatory mechanisms has led to the development of this novel method of pelvic bellows for ornithischians. This would also bolster the argument for the existence of the puboperitoneal muscle.

Additionally, it is not very clear how the other ornithischian taxa (e.g., Jeholosaurus, Stegosaurus) were measured and used in the analysis. Neither the text nor supplementary table 1 do a sufficient job explaining how these data were gathered. Furthermore, I think this table should be in the main part of the paper as it is critical to documenting what taxa were used essentially as exemplars for major clades.

There are several institutional abbreviations used in the text, including "AM" for AM 4766 numerous times. There should be an institutional abbreviation paragraph detailing those places. The same is true for supplementary table 1 in the supplementary information section.

I think the figures are very good. However, I think there can be some more detail either in the figure itself or in the text.

Figure 2: For (A) and (C) (specimen photographs), it would be helpful to denote the sternal plates. Also, there should be hyphens for SAM-PK-1332. In the description for (D), it should read SAM-PK-1332, not SAM-PK-1336.

Figure 4: The lettering in the description is mislabeled. It should be (A, B), (C, D), (E), (F), and (G).

Figures 3, 4, and 6: These figures could do well with a specimen photo for reference.

Figure 7: This is described well in the text but it feels like the associated phenograms are a bit too busy and almost detract from the salient points of the figure. Consider revising or explaining the phenograms better in the text. Or perhaps move the phenogram portion to the supplemental information section.

I appreciate Figure 8 in the discussion. You describe the comparisons to other taxa (e.g., crocodylians, avians, non-avian theropods) well in the text. I think an additional figure similar to Figure 8 illustrating these other taxa would serve your discussion very well.

---

## [Author Response]

Reviewer #2:[…] This paper has no technical or conceptual issues – as noted above, the methods are sound and well-applied, and interpretations are well-supported by the data. The paper also benefits from beautiful, very helpful figures.The only concern regards availability of the imaging (synchrotron) data. There is no mention anywhere in the text about how this can be obtained for reproducibility and future work.No additional analyses/data are needed and both the science and its presentation are strong, there is no need for improvement in these areas.My only moderate concern is there is no reference in the text that I can find on how to access arguably the most critical data within this manuscript, namely the synchrotron data of AM 4766. Either the 3D models (as STLs or 3D PDFs) or the original scan data must be made available with publication. Stating that "data are available upon request" really is no longer acceptable in such studies so dependent on medical imaging and 3D visualization.Figure 2: For (A) and (C) (specimen photographs), it would be helpful to denote the sternal plates. Also, there should be hyphens for SAM-PK-1332. In the description for (D), it should read SAM-PK-1332, not SAM-PK-1336.Figure 4: The lettering in the description is mislabeled. It should be (A, B), (C, D), (E), (F), and (G).Figures 3, 4, and 6: These figures could do well with a specimen photo for reference.Figure 7: This is described well in the text but it feels like the associated phenograms are a bit too busy and almost detract from the salient points of the figure. Consider revising or explaining the phenograms better in the text. Or perhaps move the phenogram portion to the supplemental information section.I appreciate Figure 8 in the discussion. You describe the comparisons to other taxa (e.g., crocodylians, avians, non-avian theropods) well in the text. I think an additional figure similar to Figure 8 illustrating these other taxa would serve your discussion very well.Reviewer #3:[…] While I agree with the authors' overall conclusions, it must be stated that this is still relatively speculative. The majority of the paper is describing in detail the morphology of AM 4766 and how that particular specimen (and, presumably, all Heterodontosaurus) would ventilate. The Discussion section, which is substantial, largely focuses on the evolution of the pelvic bellows. And, given the statistical tests they ran, it is a reasonable argument; however, I think more could be done with the description of the hypothetical mechanism as one moves through the ornithischian tree into other clades.Conclusion:Overall, this manuscript is a very welcome addition to the literature of varying ventilatory mechanisms in archosaurs. It is sorely needed for ornithischians as very little work has been done regarding their breathing apparatuses. More work needs to be done to investigate the appropriateness of this model in more derived clades but this paper is a good starting point for future analyses.There is not much needed for improvement of this manuscript. I would add a bit more to the comparison between other taxa and how analyses and model development for their ventilatory mechanisms has led to the development of this novel method of pelvic bellows for ornithischians. This would also bolster the argument for the existence of the puboperitoneal muscle.Additionally, it is not very clear how the other ornithischian taxa (e.g., Jeholosaurus, Stegosaurus) were measured and used in the analysis. Neither the text nor supplementary table 1 do a sufficient job explaining how these data were gathered. Furthermore, I think this table should be in the main part of the paper as it is critical to documenting what taxa were used essentially as exemplars for major clades.There are several institutional abbreviations used in the text, including "AM" for AM 4766 numerous times. There should be an institutional abbreviation paragraph detailing those places. The same is true for supplementary table 1 in the supplementary information section.I think the figures are very good. However, I think there can be some more detail either in the figure itself or in the text.Figure 2: For (A) and (C) (specimen photographs), it would be helpful to denote the sternal plates. Also, there should be hyphens for SAM-PK-1332. In the description for (D), it should read SAM-PK-1332, not SAM-PK-1336.Figure 4: The lettering in the description is mislabeled. It should be (A, B), (C, D), (E), (F), and (G).Figures 3, 4, and 6: These figures could do well with a specimen photo for reference.Figure 7: This is described well in the text but it feels like the associated phenograms are a bit too busy and almost detract from the salient points of the figure. Consider revising or explaining the phenograms better in the text. Or perhaps move the phenogram portion to the supplemental information section.I appreciate Figure 8 in the discussion. You describe the comparisons to other taxa (e.g., crocodylians, avians, non-avian theropods) well in the text. I think an additional figure similar to Figure 8 illustrating these other taxa would serve your discussion very well.

We would like to sincerely thank the reviewers of our manuscript for their generous compliments, diligent edits, and constructive comments. The changes we have made in response to their reviews are outlined below:

1. Flagged typos and syntax errors have been corrected (both in the main text, supplementary info, and their reference lists). The description for how the gastralia are arranged has been clarified.

2. Institutional abbreviations have been added to both the main text and supplementary info.

3. Segmented bones have now been labelled as such in relevant figure captions.

4. Figure 1. Directional arrow has been added to Figure 1C.

5. Figure 5. Additional abbreviations have been added to figure 5’s caption.

6. Figure 7. We have opted to follow a reviewer’s suggestion in relocating the Phenograms previously included in this figure to the supplementary information.

7. Discussion: we have rescinded our previous mention of *M. rectus abdominus* being vestigial in derived ornithischian taxa. These muscles would have indeed still been incredibly vital to these taxa in still supporting the viscera, and we have updated the text to reflect this. We thank the particular reviewer that flagged our terminology around this.

8. With respect to the scan data: these data will be available directly from the ESRF scan database.